# Single-molecule imaging reveals molecular coupling between transcription and DNA repair machinery in live cells

Han Ngoc Ho [1,2], Antoine M. van Oijen [1,2] & Harshad Ghodke [1,2 ✉]

The *Escherichia coli* transcription-repair coupling factor Mfd displaces stalled RNA polymerase and delivers the stall site to the nucleotide excision repair factors UvrAB for damage detection. Whether this handoff from RNA polymerase to UvrA occurs via the Mfd-UvrA$_2$-UvrB complex or alternate reaction intermediates in cells remains unclear. Here, we visualise Mfd in actively growing cells and determine the catalytic requirements for faithful recruitment of nucleotide excision repair proteins. We find that ATP hydrolysis by UvrA governs formation and disassembly of the Mfd-UvrA$_2$ complex. Further, Mfd-UvrA$_2$-UvrB complexes formed by UvrB mutants deficient in DNA loading and damage recognition are impaired in successful handoff. Our single-molecule dissection of interactions of Mfd with its partner proteins inside live cells shows that the dissociation of Mfd is tightly coupled to successful loading of UvrB, providing a mechanism via which loading of UvrB occurs in a strand-specific manner.

---

[1] Molecular Horizons and School of Chemistry and Molecular Bioscience, University of Wollongong, Wollongong, NSW 2522, Australia. [2] Illawarra Health and Medical Research Institute, Wollongong, NSW 2522, Australia. ✉email: harshad@uow.edu.au

DNA damage on the transcribed strand is repaired at a rate faster than lesions on the non-transcribed strand[1–3]. This enhanced transcription-coupled repair (TCR) is attributed to RNA polymerase (RNAP) acting as a damage sensor, followed by the transcription-repair coupling factor (TRCF)-dependent recruitment of the nucleotide excision repair (NER) machinery[4]. TCR is highly conserved among organisms from bacteria to humans, barring a known exception in archaea[5,6]. In several cases, TRCFs recruit the NER machinery that then probes the stall site for the presence of lesions.

In the model organism *Escherichia coli*, the prokaryotic TRCF, Mfd recognizes stalled transcriptional complexes at sites of lesions[4] and roadblocks[7,8] and orchestrates termination of transcription and recruitment of NER machinery (UvrAB) to the site. Mfd is a modular, multi-domain protein and functions as an adapter protein that couples RNAP and the NER proteins (Fig. 1a)[9]. Within its N-terminus, domains D1a, D2 and D1b constitute the UvrB homology module (BHM)—a structural homolog of UvrB that is capable of interactions with UvrA[9]. Domain D3 has an as yet unknown function. Domain D4, the RNAP interacting domain (RID), interacts with the β subunit of RNAP[9,10]. Domains D5/D6 together constitute the motor domains of Mfd bearing homology to the superfamily 2 helicase RecG[9,11]. This motor domain orchestrates translocation of Mfd in the 3′–5′ direction with respect to the transcribed strand[12–14]. Finally, domain D7 interacts with the N-terminus to lock the protein in an auto-inhibited conformation that is unable to stably engage DNA in the absence of interactions with RNAP[9,11,15–17].

Mfd is recruited to the upstream edge of a stalled ternary elongation complex (TEC)[10]. During TCR, this recruitment is accompanied by a release of Mfd's auto-inhibition leading to the activation of its translocase activity that eventually displaces RNAP, and a concomitant exposure of the UvrB homology module (BHM) to solution[9,13,15,17,18]. Next, DNA-bound and RNAP-engaged Mfd recruits the UvrAB proteins to the site[4] via the BHM. In vitro studies indicate that UvrA (and to a greater extent UvrAB) enhances the dissociation of Mfd-RNAP from the DNA[19] leading to the formation of the pre-incision complex by UvrB on DNA (Fig. 1b).

However, TCR is not the only outcome of interactions of Mfd with RNAP. Mfd can also reactivate transcription at class II pause sites where RNAP is backtracked in a manner in which the catalytic site of RNAP is occluded by RNA[10,12,20]. In vitro single-molecule studies using optical traps revealed that Mfd can translocate on forked DNA substrates anchored between a bead and a coverslip[12]. Further, dual optical trap experiments revealed a 'release and catch-up' mechanism, where a single translocating Mfd can reactivate a nucleotide-starved stalled TEC in the presence of rNTPs, and continue to translocate on the DNA until it re-encounters RNAP at the next stall/pause site[12]. Currently, no experimental evidence is available to suggest that the NER factors are recruited to these sites.

To understand how Mfd interacts with RNAP and downstream repair factors UvrAB in its physiological context inside living *Escherichia coli*, we created a fluorescently labeled Mfd-YPet fusion that retains wild-type functions[21]. Consistent with in vitro

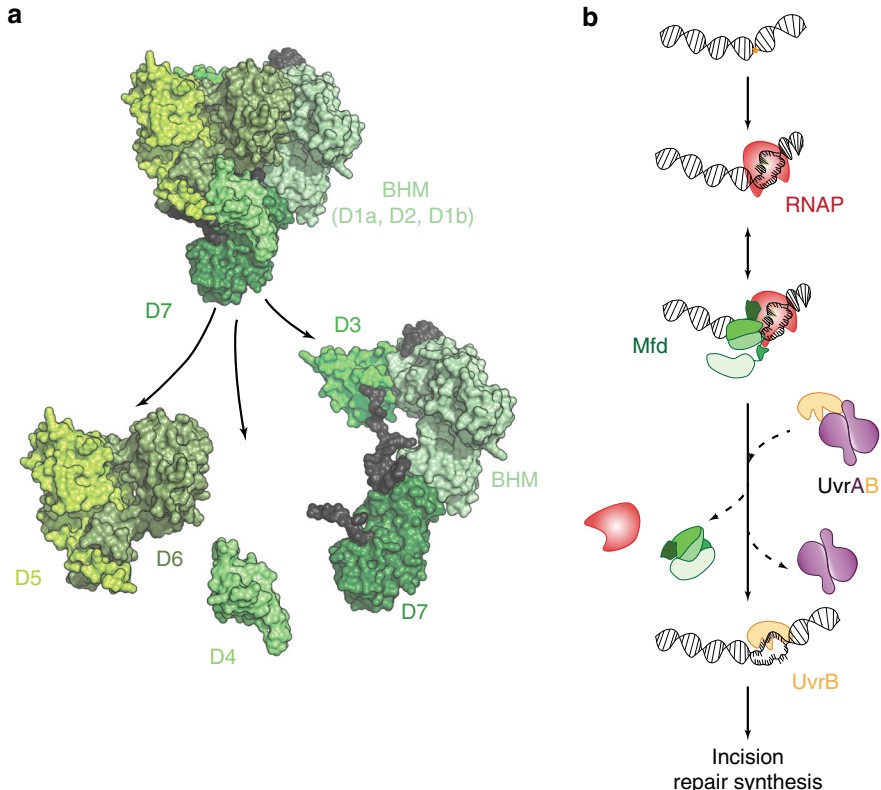

**Fig. 1 In vitro model for the initial stages of transcription-coupled repair. a** Apo-structure of Mfd (PDB ID: 2EYQ)[9]. The modular Mfd protein consists of seven domains connected by three linkers. The N-terminus (D1a, D2, D1b) is structurally homologous to UvrB and constitutes the UvrB-homology module (BHM). The BHM binds the auto-inhibitory C-terminus (D7) resulting in packing of the RNA polymerase interacting domain (RID; domain 4) against the translocation module that consists of a superfamily 2 helicase (SF2) motor (domains D5-D6). Engagement of stressed transcriptional complexes occurs via RID, resulting in loss of inhibition (reviewed in ref. [54]). **b** Ternary elongation complexes stall upon encountering DNA damage in the template strand. According to in vitro models, Mfd engages stalled RNAP and recruits UvrA and UvrB to the site followed by loading of UvrB on the DNA[4,19]. The exact sequence of events in vivo is unclear.

findings[15–18], we found that Mfd is auto-inhibited in cells[21], exhibiting extremely transient DNA binding in the absence of interactions with RNAP. Further, interactions with RNAP are enriched in the presence of a drug that stalls RNAP (CBR703[22]), and absent in cells treated with the transcription inhibitor rifampicin. The abundance of freely diffusive cytosolic Mfd as well as Mfd molecules stably interacting with RNAP revealed that Mfd only engages a fraction of transcribing RNAP during normal growth. This finding is consistent with extensive in vitro data characterizing Mfd's substrates: stalled RNAP at sites of lesions[4,14,19], roadblocks[7] and backtracked RNAP at class II pause sites[14]. Since Mfd is not known to interact with TECs that are actively executing the nucleic acid addition cycle, we suggested that these in vivo observations reflected interactions with stressed TECs that are paused or stalled in manners described above, potentially at sites of endogenous lesions[21]. Our quantification of binding lifetimes of Mfd revealed a complete dependence on the presence of UvrA. In cells lacking *uvrA*, the lifetime of Mfd was longer (~29 s) compared to that in *uvrA+* cells (~18 s), consistent with measurements from experiments targeted at a single-molecule in vitro reconstitution of TCR at sites of lesions[19]. The abundant observations of RNAP- and UvrA-dependent interactions of Mfd observed in normal growing cells in our experiments agree with arguments that TCR represents a house-keeping DNA repair pathway[19,23]. When exposed to genotoxins, bacterial cells trigger the SOS response and rapidly increase the expression of NER factors[24–28]. Since UvrAB are able to search for and trigger repair on chromosomal DNA independently of Mfd[29], the signal from TCR becomes difficult to observe after SOS induction[4,30]. For these reasons, it has additionally been argued that observation of TCR should optimally be performed in non-SOS induced conditions[23].

Important questions on the nature of interactions of Mfd with RNAP and the UvrAB proteins inside live cells are currently unanswered: what is the identity of the hand-off complex formed at the nexus of transcriptional stress and NER in the physiological context? What catalytic activities are required for its formation and resolution? Using a method to reliably measure binding lifetimes of DNA-bound repair factors in cells[21,24,31], here we measure the DNA-binding lifetime of Mfd in the presence of ATPase mutants of UvrA and damage recognition mutants of UvrB in normal growing cells. We find that in cells expressing mutants of either UvrA or UvrB, Mfd is stably arrested on DNA compared to wild-type cells. Our results demonstrate that in wild-type cells, ATP hydrolysis by UvrA and stable loading of UvrB greatly enhance the rate of dissociation of Mfd. Combined with the observations that UvrA is arrested in cells lacking UvrB[32], these findings lead us to propose a 'facilitated dissociation' model for interactions between the transcription repair coupling factor and the repair machineries in cells. We propose that the UvrA-mediated loading of UvrB on DNA facilitates the dissociation of Mfd and UvrA, and completes the handoff. This model provides an elegant mechanism for the strand- and site-specific loading of the repair machinery at the stall site.

## Results

**Imaging of Mfd in live cells**. We have previously created and validated a chromosomally expressed fusion of Mfd to the yellow fluorescent protein (YPet) that is functional in TCR, and that is recruited to DNA via paused/stalled TECs with a bound lifetime of $18 \pm 1$ s (mean ± standard deviation of bootstrap distribution) in wild-type cells[21]. We also found that in the absence of UvrA, the lifetime ($\tau_{\text{Mfd}|\Delta uvrA}$) of the interaction of Mfd with stalled/paused RNAP on DNA increases to $29 \pm 2$ s[21]. Having established that UvrA is necessary for dissociating Mfd with a lifetime of 18 s,

we next investigated the role of UvrB in promoting the dissociation of Mfd.

To that end, we measured the lifetime of DNA-bound Mfd-YPet in cells lacking the downstream repair factor UvrB. First, we created a strain that expresses Mfd-YPet from the native chromosomal *mfd* locus while lacking the *uvrB* gene (*mfd-YPet ΔuvrB*). Next, we immobilised these cells on a modified glass coverslip in a flow cell such that exponentially growing cells could be reliably imaged for several hours in transparent rich growth medium (Fig. 2a). The fluorescence of Mfd-YPet molecules manifested as a mixture of static foci arising from DNA-bound molecules, and cytosolic background arising from diffusive molecules (Fig. 2b). We first collected videos of cells expressing Mfd-YPet *ΔuvrB* with continuous exposure times of 0.1 s (Fig. 2c, Supplementary Movie 1).

Observation of long-lived binding events of fluorescently tagged proteins is limited by the photobleaching of the fluorescent proteins upon continuous exposure to excitation photons. To extend the observation time window, we adopted interval imaging in which consecutive exposures were spaced out by the addition of a dark interval of duration $\tau_d$ (see ref. [24,31] and Methods). We collected a series of videos each with a unique dark interval and measured the apparent binding lifetimes of single Mfd-YPet molecules. For each $\tau_d$ condition, this allows the generation of dwell time distributions. Fitting the dwell time distribution yields the effective rate of focus loss, reflecting a sum of the photobleaching rate of the fluorescent protein and dissociation rate of Mfd-YPet (Fig. 2e, solid line). We used a large set of distinct $\tau_d$ spanning three orders of magnitude (0.1–9.9 s) to ensure accurate measurements of binding lifetimes that last for seconds to several minutes inside cells (Fig. 2d, e). Finally, since the photobleaching rate was maintained constant across the various acquisitions, we extracted binding lifetimes by global fitting of the entire data set to single- or bi-exponential models[24].

In cells lacking UvrB, the dissociation of Mfd-YPet is well described by a single-exponential function with a lifetime ($\tau_{\text{Mfd}|\Delta uvrB}$) of $139 \pm 20$ s (mean ± standard deviation of bootstrap distribution; Table 1, Supplementary Fig. 1). This lifetime is five times longer than in cells lacking UvrA (29 s), and eight times longer than in wild-type cells (18 s)[21]. Further, the lifetime of $139 \pm 20$ s is similar to that of UvrA-YPet in *uvrA-YPet ΔuvrB* cells ($97 \pm 18$ s)[32]. Taken together, these results suggest that a highly stable DNA-bound Mfd-UvrA$_2$ complex is formed in the absence of UvrB (Fig. 2f).

We then attempted to create *mfd-YPet* cells that co-express UvrA-PAmCherry with the objective of detecting the long-lived Mfd-UvrA using dual color imaging. However, our attempts to create UvrA-RFP constructs (either PAmCherry or mKate2) resulted in poorly expressed proteins or truncated gene products ultimately rendering this strategy unviable.

**The distal ATPase of UvrA governs stable association with Mfd**. We next investigated key catalytic properties of UvrA that regulate the assembly and disassembly of the Mfd-UvrA$_2$ complex. Since engagement of UvrA with UvrB and DNA is regulated by nucleotide binding and hydrolysis at the two ATPase sites of UvrA[33–36], we hypothesized that interactions with Mfd may also be governed by ATP hydrolysis. To assess the role of ATP hydrolysis in the formation of the Mfd-UvrA$_2$ intermediate, we engineered strains that express either UvrA(K37A) (proximal ATPase mutant) or UvrA(K646A) (distal ATPase mutant) from the native *uvrA* locus in both *mfd-YPet* and *mfd-YPet ΔuvrB* backgrounds (Fig. 3a; Supplementary Methods).

First, we measured the DNA-bound lifetimes of Mfd-YPet in cells expressing the UvrA(K646A) mutant. This mutant is severely

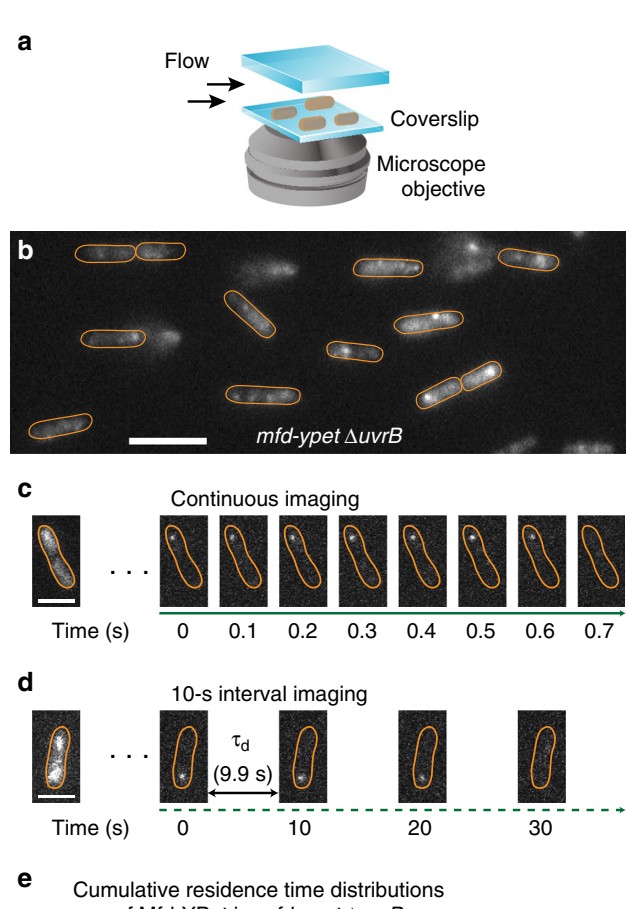

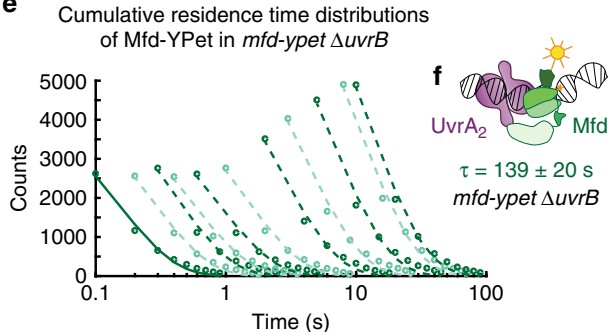

**Fig. 2 Measurements of Mfd-YPet binding lifetimes in cells lacking UvrB.**
**a** Cells expressing fluorescent Mfd-YPet were grown to early exponential phase and loaded in a flow cell. Cells were imaged under constant supply of aerated growth medium for several hours. **b** Representative fluorescence image of *mfd-YPet ΔuvrB* cells upon exposure to 514-nm light. Scale bar, 5 µm. Cell outlines (orange) are provided as a guide to the eye. **c** Fluorescence data were collected in video format, with 0.1-s frames taken continuously. Due to fast photobleaching of the fluorescent protein YPet, continuous imaging hinders the observation of binding events on the second timescale. Scale bar, 2 µm. **d** To extend the observation time window up to several minutes, images can be acquired with a dark interval ($\tau_d$) inserted between consecutive 0.1-s frames. Scale bar, 2 µm. **e** Cumulative residence time distributions (CRTDs; circles) of Mfd-YPet in *mfd-YPet ΔuvrB* cells and the corresponding single-exponential fits (lines) obtained from continuous imaging (solid line) or interval imaging (dashed lines) with a dark interval ($\tau_d$) between consecutive frames. $\tau_d$ increases from 0.1 s to 9.9 s (left to right, see Methods). CRTDs are constructed from eight biologically independent repeats ($N_{repeats}$) totaling $n_{obs} = 37,682$ individual observations (also see source data file). **f** Global fitting of CRTDs in (**e**) yields a lifetime of 139 ± 20 s, suggesting Mfd (green) is arrested in complex with UvrA₂ (purple) in cells lacking UvrB. Error bars are standard deviations from ten bootstrapped CRTDs. Source data are provided as a Source Data file.

defective in NER and TCR since it can load UvrB to only 1% of the level of wild-type UvrA[13,33]. In the presence of UvrA(K646A), we detected two kinetic sub-populations of Mfd-YPet in cells lacking UvrB: a population exhibiting a short lifetime of 1.1 ± 0.1 (68 ± 2%) and another with a long lifetime ($\tau_{Mfd|\ uvrA(K646A)\ \Delta uvrB}$) of 26 ± 2 s (32 ± 2%) (Fig. 3b, Table 1, Supplementary Fig. 2a, b). The fast dissociating sub-population (1.1 ± 0.1 s) may represent non-specific binding of Mfd-YPet to DNA[21,37]. In the presence of UvrB, Mfd dissociated with a single lifetime ($\tau_{Mfd|\ uvrA(K646A)}$) of 37 ± 3 s in cells carrying the UvrA(K646A) mutant (Fig. 3b; Table 1, Supplementary Fig. 3c, d). The long lifetimes measured here are comparable to that of Mfd-YPet in cells lacking UvrA (29 ± 2 s)[21]. Notably, the absence of UvrB did not lead to an arrest of Mfd-YPet on the DNA in cells carrying UvrA(K646A) as it did in the case of Δ*uvrB* cells expressing wild-type UvrA (139 ± 20 s). A simple interpretation of these results is that the K646A mutant of UvrA is unable to engage RNAP-bound Mfd stably. A similar interpretation has been drawn in the case of the formation of the UvrAB complex[38]. This interpretation is also consistent with the report that UvrA(K646A) engages UvrB poorly[33].

**The proximal ATPase of UvrA regulates dissociation of Mfd.** We then investigated the influence of the proximal ATPase on the lifetime of the Mfd-UvrA complex. In *mfd-YPet uvrA(K37A) ΔuvrB* cells, we measured the lifetime of Mfd-YPet ($\tau_{Mfd|\ uvrA(K37A)\ \Delta uvrB}$) to be 304 ± 69 s (Fig. 3b, Table 1, Supplementary Fig. 3a, b), tenfold longer than that of Mfd-YPet in Δ*uvrA* cells ($\tau_{Mfd|\Delta uvrA} = 29 \pm 2$ s)[21], and almost twofold longer than that in Δ*uvrB* cells (139 ± 20 s). In *mfd-YPet uvrA(K37A)* cells that express UvrB, the lifetime of Mfd-YPet ($\tau_{Mfd|\ uvrA(K37A)}$) was 52 ± 4 s (Fig. 3b, Table 1, Supplementary Fig. 3c, d). It has been previously shown that UvrA (K37A) can load UvrB on DNA with 10% efficiency compared to wild-type UvrA[33] and can support the preferential repair of the damaged template strand in vitro[13]. To reconcile our observations with these findings, we propose that the apparently single population detected in our experiments represents a mixture of two populations with unresolvable lifetimes (within the size of the collected data set)—one that can potentially load UvrB on DNA, and a population that is unable to load UvrB efficiently (Fig. 3e, f). Notably, unlike UvrA(K646A) which fails to stably engage Mfd, UvrA(K37A) can arrest Mfd on DNA. The disassembly of this Mfd-UvrA(K37A)₂ complex is greatly aided by UvrB.

**UvrB mutants arrest Mfd on DNA.** UvrA loads UvrB at the site of DNA damage during NER. In this step, single-stranded DNA is threaded in a cleft formed by the absolutely critical β-hairpin of UvrB[39,40] (Fig. 4a, b), followed by interrogation of the nucleobases mediated by UvrB's cryptic helicase activity[41,42]. Since loading of UvrB promotes dissociation of UvrA from the damage surveillance complex[32,43], we next investigated whether dissociation of Mfd from the handoff complex is also regulated by loading of UvrB on DNA. We measured the residence time of Mfd-YPet in cells expressing two β-hairpin mutants of UvrB from the native chromosomal locus[39,41,44] (Fig. 4c, Supplementary Methods). In cells expressing UvrB molecules lacking the β-hairpin (UvrB(ΔβHG)), the lifetime of Mfd ($\tau_{Mfd|\ uvrB(\Delta\beta HG)}$) was found to be 188 ± 46 s—comparable to that in cells lacking UvrB (139 ± 20 s), and an order of magnitude longer than in the wild-type UvrB background (Fig. 4d, Table 1, Supplementary Fig. 4a, b). We then repeated these experiments in cells expressing the Y96A mutant of UvrB that can be loaded on DNA but fails to support damage verification[40,44]. The lifetime of Mfd ($\tau_{Mfd|\ uvrB(Y96A)}$) was found to be 70 ± 12 s in this background (Fig. 4d, f, Table 1, Supplementary Fig. 4c, d). Considering that both UvrB mutants retain the ability to form UvrAB-DNA complexes[41,44], the

**Table 1 Lifetimes of Mfd-YPet in various genetic backgrounds.**

| Derivatives of *mfd-YPet* | Slow lifetime (s) (mean ± s.d.) | Percentage of the slow population (mean ± s.d.) | Fast lifetime (s) (mean ± s.d.) | N repeats |
|---|---|---|---|---|
| *mfd-YPet*[21] | 18 ± 1 | 100 | – | 9 |
| Δ*uvrA*[21] | 29 ± 2 | 37 ± 4 | 0.5 ± 0.04 | 10 |
| Δ*uvrB* | 139 ± 20 | 100 | – | 8 |
| *uvrA(K646A)* Δ*uvrB* | 26 ± 2 | 32 ± 2 | 1.1 ± 0.1 | 6 |
| *uvrA(K646A)* | 37 ± 3 | 100 | – | 11 |
| *uvrA(K37A)* Δ*uvrB* | 304 ± 69 | 100 | – | 7 |
| *uvrA(K37A)* | 52 ± 4 | 100 | – | 8 |
| *uvrB(ΔβHG)* | 188 ± 46 | 100 | – | 9 |
| *uvrB(Y96A)* | 70 ± 12 | 100 | – | 8 |
| Δ*uvrB* /pUvrB | 11.1 ± 0.7 | 26 ± 2 | 0.50 ± 0.08 | 6 |

simplest explanation is that the handoff complexes formed by Mfd-UvrA₂ and mutants of UvrB are impaired in evicting Mfd (Fig. 4e, f). While the lifetime of Mfd-YPet in *uvrB(Y96A)* cells is nearly four-fold longer than that of Mfd-YPet in wild-type UvrB (70 s *vs.* 18 s, Table 1), Mfd-YPet in *uvrB(Y96A)* still dissociates faster when compared to *uvrB(ΔβHG)* and Δ*uvrB* backgrounds. These data suggest UvrB(Y96A), while being inefficient, can still catalyse dissociation of Mfd in Mfd-UvrA complex. The discrepancy between UvrB(Y96A) and UvrB(ΔβHG) observed in our experiments is unexpected, considering their similar abilities to bind UvrA and the inability to form stable post-incision UvrB-DNA complexes[41,44]. Unlike UvrB(ΔβHG), UvrB(Y96A) possesses hydrophobic residues at the tip that can destabilize dsDNA. Perhaps this disruption plays a role in promoting Mfd dissociation.

These results have three important implications: (1) since Mfd and UvrB do not physically interact[4,19], the stabilisation of Mfd on DNA in mutant UvrB backgrounds must occur via the Mfd-UvrA₂-UvrB handoff complex, (2) since mutants of UvrB can arrest Mfd on DNA, we infer that UvrB does not simply compete off Mfd from UvrA₂ complexes at the binding interface occupied by Mfd and (3) binding of UvrB to the handoff complex is not sufficient for eviction of Mfd and UvrA from the handoff complex, successful engagement of UvrB with DNA is a necessary requirement.

**UvrB levels determine the rate of dissociation of Mfd.** Since DNA-repair enzymes must navigate the problem of target search, i.e., locating sparse DNA damage in a vast excess of undamaged DNA, we next investigated whether the diffusion of UvrB to the site of the Mfd-UvrA₂ complex limits the handoff rate. Elevated copy numbers of UvrB achieved by expressing UvrB from its native promoter maintained on a low-copy plasmid (3–4 copies per cell[45]) resulted in faster dissociation of Mfd (Fig. 4d, g; Table 1, Supplementary Fig. 4e, f). In this case, the lifetime of Mfd ($\tau_{Mfd|~UvrB\uparrow\uparrow}$) was 11.1 ± 0.7 s, shorter than that of Mfd-YPet in the wild-type levels of UvrB (18 ± 1 s)[21]. The observation that modest overexpression of UvrB can reduce the binding lifetime of Mfd from 18 s to 11 s, suggests that the observed residence time of Mfd-YPet reflects at least two processes: first, a wait time in which Mfd awaits the arrival of UvrAB, and a second process that encapsulates activities downstream of UvrAB recruitment such as ATP hydrolysis mediated loading of UvrB. Consistent with this interpretation, Mfd-YPet is turned over faster in UV-treated cells that express high amounts of UvrB after SOS induction[32].

## Discussion
Faithful handoff of DNA containing lesions to downstream factors is an inherent challenge faced by DNA-repair machineries[46].

In this work, we demonstrate that in live bacterial cells dissociation of Mfd requires ATP hydrolysis at the proximal site of UvrA, and a UvrB protein with an intact β-hairpin motif (Fig. 5). The implication that directional loading of the damage verification machinery serves as a molecular switch to trigger the dissociation of upstream damage recognition factors provides an elegant solution to the challenge of faithful handover, and may be a universally conserved feature from prokaryotes to eukaryotes.

Our previous study revealed that the lifetime of *bona fide* interactions of Mfd with RNAP in growing cells is 18 s (Fig. 5a)[21]. Unexpectedly, in cells disabled in their ability to perform TCR owing to a deletion of *uvrA*, Mfd exhibited a binding lifetime of 29 s (Fig. 5a)[21]. Since Mfd only interacts with DNA non-specifically (in the absence of RNAP interactions) on the time-scale of a few hundred milliseconds[21], this enhanced lifetime of 29 s must arise from an interaction after engagement with RNAP. Clues to Mfd activity during this 29 s time window may be found in in vitro work indicating that after remodeling RNAP, Mfd remains bound to RNAP and continues to translocate on naked DNA templates for several kilo base pairs in vitro[14,19,47,48]. This continued translocation of Mfd can promote repair of lesions downstream of the RNAP stall/pause site[14]. The enhanced repair of downstream lesions is reduced in the presence of DNA binding proteins on the substrate[14] suggesting that the processive translocation of Mfd can be interrupted by DNA binding proteins occupying the template. In cells, we suggest that the processivity (and consequently lifetime) of translocating Mfd-RNAP complex could be truncated by encounters with DNA organizing proteins as well as replication, recombination and transcription machineries that normally occupy chromosomal DNA. Assuming a translocation rate of ~4 nucleotides per second[48] in cells lacking UvrA, Mfd (residence time of 29 s) can travel an average distance of 116 nucleotides downstream of the site of recruitment—a length scale on which Mfd promotes repair of downstream lesions in vitro[14].

Mfd binds DNA very stably in *uvrA*⁺ cells lacking *uvrB* (139 ± 20 s) (Fig. 5a). Strikingly, UvrA-YPet is also retained on DNA with a similar lifetime of 97 ± 18 s in *mfd*⁺ cells lacking *uvrB*[32]. The corroboration of these two measurements and the finding that UvrB promotes the dissociation of both Mfd and UvrA suggests that they form a Mfd-UvrA complex in cells. Consistent with previous observations[19], our live cell study confirms that the Mfd-UvrA complex is formed at the intersection of the transcription and repair pathways. In cells, various DNA processing enzymes acting on the DNA may result in the concomitant dissociation of both Mfd, and UvrA from DNA in the absence of UvrB.

Importantly, UvrB (but not loading-deficient UvrB mutants) facilitates the dissociation of Mfd. Considering that UvrB does not interact directly with Mfd[4], the facilitated dissociation of Mfd

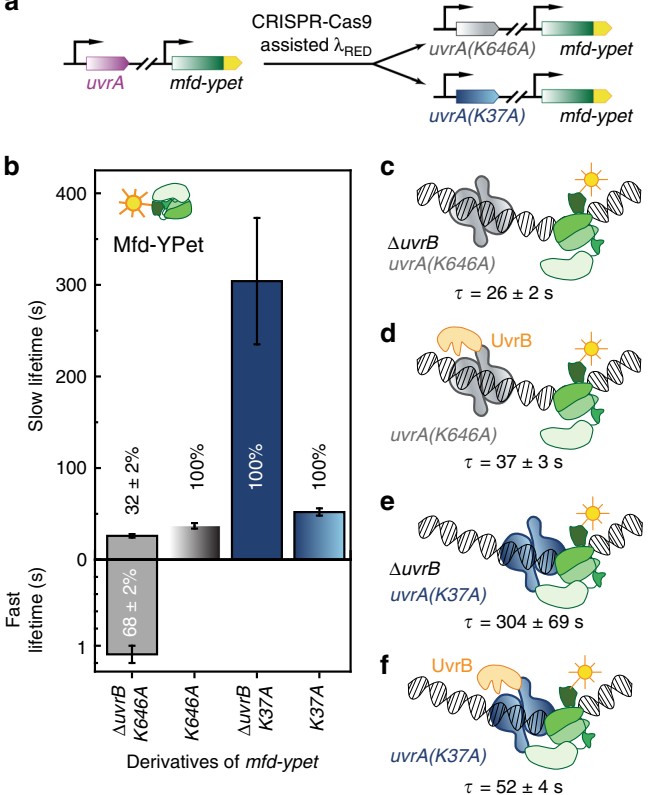

**Fig. 3 Measurements of Mfd-YPet binding lifetimes in cells expressing mutant UvrA deficient in ATP binding and hydrolysis. a** The wild-type *uvrA* allele was replaced by the mutant allele in *mfd-YPet* cells, so that mutant cells either express the distal ATPase mutant UvrA(K646A) or the proximal ATPase mutant UvrA(K37A) from the chromosome. **b** Bar plots show lifetimes of DNA-bound Mfd-YPet in the corresponding genetic backgrounds. Where two kinetic sub-populations are detected, the fast lifetime is displayed in the lower panel. Numbers in percentages represent the amplitudes of kinetic sub-populations (see Supplementary Figs. 2, 3). **c** Lifetime of Mfd-YPet in *mfd-YPet uvrA(K646A)* Δ*uvrB* (26 ± 2 s; $N_{repeats}$ = 6 totaling $n_{obs}$ = 23,763 individual observations) is similar to that of Mfd-YPet in Δ*uvrA* cells (29 ± 2 s[21]), suggesting the distal ATPase mutant (gray) is unable to interact with Mfd (green). **d** Lifetime of Mfd-YPet in the presence of UvrA(K646A) and UvrB (orange) (37 ± 3 s; $N_{repeats}$ = 11 totaling $n_{obs}$ = 36,073 individual observations) also resembles that of Mfd-YPet in Δ*uvrA* cells. **e** Mfd-YPet is arrested in the presence of the proximal ATPase mutant UvrA(K37A) in cells lacking UvrB (lifetime of 304 ± 69 s; $N_{repeats}$ = 7 totaling $n_{obs}$ = 45,254 individual observations). **f** Mfd-YPet dissociates in 52 ± 4 s ($N_{repeats}$ = 8 totaling $n_{obs}$ = 36,486 individual observations) in the presence of UvrA(K37A) and UvrB, suggesting that ATP hydrolysis at the proximal site is required for promoting the dissociation of Mfd-YPet. Error bars are standard deviations from ten bootstrapped CRTDs. Source data are provided as a Source Data file.

observed here suggests that UvrB exerts this influence via UvrA in a transient Mfd-UvrA₂-UvrB handoff complex that has been previously proposed[4,14,19]. Whereas the formation of this complex requires ATP hydrolysis at UvrA's distal ATPase, we found that ATPase activity of UvrA at the proximal site is necessary for dissociation of Mfd with a lifetime of 18 s in wild-type cells. Since Mfd is not indefinitely arrested in UvrA(K37A) cells, but instead exhibits a tenfold longer residence time we infer that ATP hydrolysis serves to enhance the rate of dissociation and is not strictly required for complex disassembly. Nevertheless, the observation that the presence of UvrB partially alleviates the

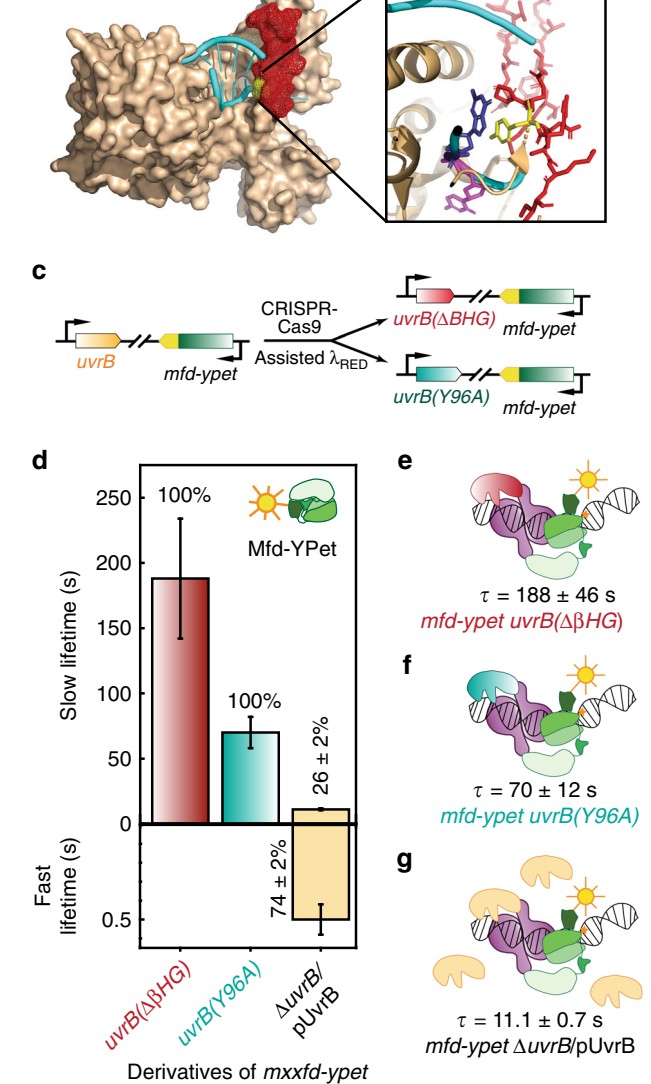

**Fig. 4 Measurements of Mfd-YPet binding lifetimes in cells expressing mutant UvrB deficient in DNA loading or over-expressing wild-type UvrB. a** Crystal structure of *Bacillus caldotenax* UvrB (brown) bound to a DNA hairpin (cyan) (PDB ID: 2FDC)[39]. The β-hairpin of UvrB is highlighted in red. **b** Zoomed-in view of the tyrosine residue Y96 (yellow) at the base of the β-hairpin (red). The tyrosine residue stabilises the DNA by forming π-stacking interactions with a guanine (blue) while a neighboring base (magenta) is flipped out into a hydrophobic pocket of UvrB. **c** Schematic of CRISPR-Cas9 assisted incorporation of deletions and point mutations in the *uvrB* gene. **d** Bar plots show lifetimes of DNA-bound Mfd-YPet in the corresponding genetic backgrounds. Where two kinetic sub-populations are detected, the fast lifetime is displayed in the lower panel. Numbers in percentages represent the amplitudes of kinetic sub-populations (see Supplementary Fig. 4). **e** Lifetime of Mfd-YPet in cells expressing mutant UvrB lacking the β-hairpin (red) was found to be 188 ± 46 s ($N_{repeats}$ = 9 totaling $n_{obs}$ = 37,950 individual observations), demonstrating this mutant UvrB fails to promote the dissociation of Mfd-YPet (green). **f** Similarly, mutant UvrB in which the absolutely conserved residue Y96 is replaced with alanine (cyan) also fails to promote the dissociation of Mfd-YPet ($\tau_{Mfd| uvrB(Y96A)}$ = 70 ± 12 s; $N_{repeats}$ = 8 totaling $n_{obs}$ = 35,082 individual observations). **g** Overexpression of UvrB (orange) leads to a shorter lifetime of DNA-bound Mfd-YPet, compared to the constitutive level of UvrB (11.1 ± 0.7 s vs. 18 ± 1 s[21]). $N_{repeats}$ = 6 totaling $n_{obs}$ = 20,653 individual observations. Error bars are standard deviations from ten bootstrapped CRTDs. Source data are provided as a Source Data file.

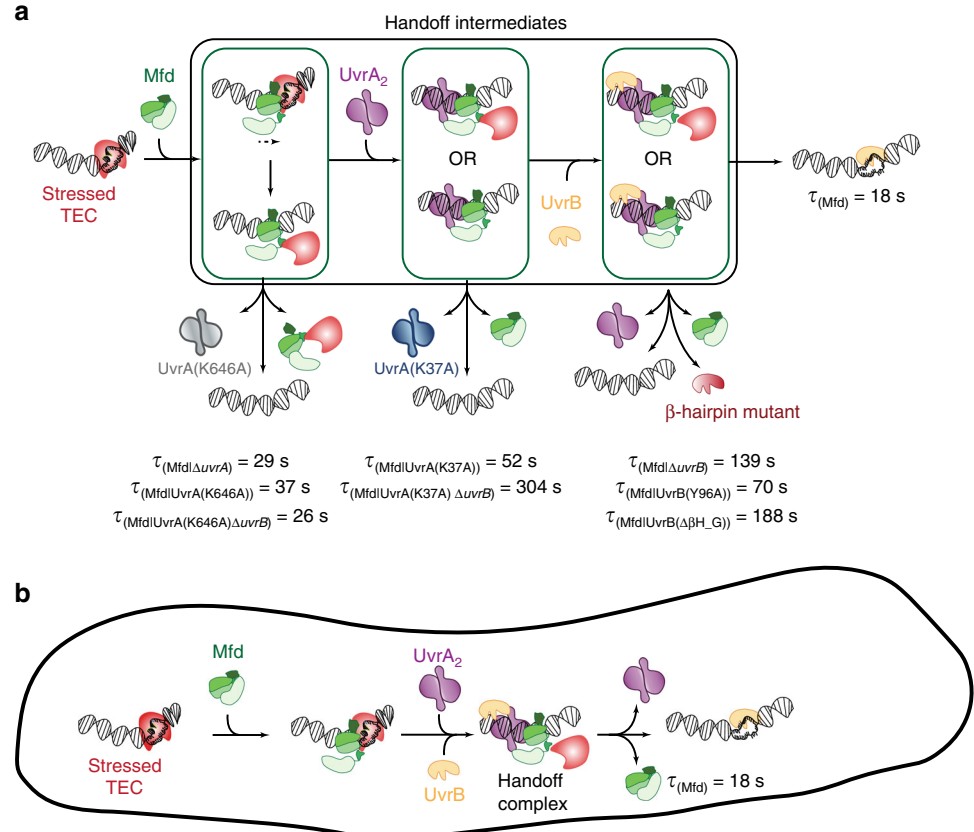

**Fig. 5 Model for interactions of Mfd, UvrA and UvrB at sites of stressed TECs in cells. a** In cells, Mfd is normally auto-inhibited and does not stably interact with DNA. Binding to stressed TECs via its RNAP interacting domain leads to release of inhibition and a large conformational change in the protein. Following this, the motor domain engages DNA on the upstream edge of the transcription bubble and subsequent ATP hydrolysis leads to either transcription reactivation/termination. Mfd continues to translocate on dsDNA after remodeling the RNAP. Meanwhile, the UvrB homology module of Mfd is exposed permitting recruitment of UvrA. ATP hydrolysis at UvrA's distal ATPase site is necessary for this engagement. Engagement with $UvrA_2$ leads to arrest of translocation. Further, recruitment of UvrB (either sequentially as shown, or simultaneously with $UvrA_2$) to the site leads to formation of the Mfd-UvrA-UvrA-UvrB handoff complex. Subsequent ATP hydrolysis at UvrA's proximal ATPase site results in loading of UvrB on the DNA. Successful engagement of UvrB with the DNA via its β-hairpin is necessary for completing the handoff, culminating in dissociation of Mfd and $UvrA_2$ from the site. **b** Facilitated dissociation model operates inside cells resulting in the simultaneous loss of $UvrA_2$ and Mfd from the site of transcriptional stress.

deficiencies of the UvrA(K37A) mutant suggests that this ATPase activity and the presence of an intact β-hairpin required for loading of UvrB on DNA together greatly enhance the rate of dissociation of Mfd. We reach this conclusion based on the observation that β-hairpin mutants of UvrB that are impaired in their ability to form a stable UvrB-DNA complex[41,44] when expressed from the native chromosomal locus, delay the dissociation of Mfd from DNA. Since these mutant UvrB proteins are unaffected in their ability to interact with UvrA[41,44], the mere binding of UvrB to Mfd-$UvrA_2$ is not a sufficient allosteric signal that triggers Mfd dissociation. Instead, a downstream activity, potentially the insertion of the β-hairpin between the two DNA strands constitutes the critical step for disruption of the $UvrA_2$-UvrB interface in the Mfd-$UvrA_2$-UvrB handoff complex. Since loading of UvrB promotes dissociation of $UvrA$[4,32,43] and Mfd from Mfd-$UvrA_2$-UvrB handoff complexes, these findings demonstrate the operation of a UvrB-facilitated dissociation model describing the disassembly of the handoff complex inside cells (Fig. 5b). Consistent with this expectation, Mfd and UvrA exhibit DNA binding identical lifetimes in UV-treated cells[32], a condition in which the repair of cyclobutane pyrimidine dimers via TCR is prioritized[30].

How might the handoff complex orchestrate strand-specific loading of UvrB? Mfd is a 3′→5′ translocase that occupies and tracks the template strand[11,14]. Whereas Mfd itself is not known

to recognize the DNA damage, this strand polarity of Mfd marks the identity of the damaged strand in cases where RNAP may be stalled by endogenous lesions. In the facilitated dissociation model presented here, we envision that the arrested Mfd-$UvrA_2$ complex upon engagement with UvrB undergoes ATP hydrolysis at the proximal site leading to large conformational changes[49,50]. These could place UvrB proximal to the partially destabilized dsDNA being probed by $UvrA_2$[50,51]. An intact β-hairpin that is essential for separation of the two strands and for stable engagement with the parted dsDNA, would then permit directional loading of UvrB at the stall site. The details of the molecular architectures of the DNA-bound Mfd-$UvrA_2$ and Mfd-$UvrA_2$-UvrB sub-complexes described here remain unavailable, but it is tempting to speculate that the Mfd-$UvrA_2$-UvrB handoff complex that contacts DNA through each of Mfd, $UvrA_2$, and UvrB might be highly strained. The resolution of such a strained complex may first occur via the disruption of the Mfd-DNA interface (which would represent the weakest point of contact since its affinity is modulated by the hydrolysis state of the ATP; $K_D$ for this affinity has been shown to be low nM when bound to ATPγS[12] and is expected to be significantly weaker in the presence of ADP) followed by disruption of the $UvrA_2$-dsDNA interface ($K_D$ is in the 1–10 nM range depending on the presence/absence of DNA damage[52,53]). Since the Mfd-UvrA interaction is tight (low pM binding affinity[19]) and the $UvrA_2$-UvrB interaction

is comparatively weaker (~10 nM $K_D$[52]), we expect that $UvrA_2$ would dissociate from UvrB before it dissociates from Mfd.

Our findings have important implications for how TCR might occur in cells. Mfd can interact with paused TECs and those stalled at sites of lesions and roadblocks. Considering that the Mfd-YPet construct supports TCR in cells[21], a subset of the observations in our experiments may arise from interactions with lesion-stalled TECs. Since quantifying the amount of endogenous DNA damage in the bacterial cells in our setup is currently infeasible, estimating the fraction of events at sites of RNAP stalling lesions is challenging. Nevertheless, the observation that the entire detectable population of Mfd is arrested on DNA in cells lacking UvrB suggests that the facilitated dissociation observed here must operate even in cases where Mfd interacts with lesion-stalled TECs. A further implication of the simultaneous dissociation of Mfd and UvrA is that UvrB must be loaded in a strand-specific manner. UvrB dependent dissociation of Mfd provides an elegant explanation for the observed preferential repair of the transcribed strand. We note that the experiments presented in this study do not permit us to comment on the extent to which the observed interactions occur at sites of endogenous lesions, and whether repair indeed occurs at these sites. These ideas await further investigation.

These studies raise larger questions on the coupling of stressed TECs and the repair machinery, and the role of Mfd in mediating such coupling. Currently, no evidence is available to suggest that Mfd possesses the ability to detect damage, suggesting that Mfd is agnostic to the presence of DNA damage. Indeed, all data point to RNAP being the damage sensor. Instead, the observations that Mfd can recognize various types of stressed TECs (paused RNAP, and those stalled at lesions or protein roadblocks) indicate that Mfd functions as a sensor for transcriptional stress. Our findings force us to ask the question: do the different types of stressed TECs elicit different responses from Mfd and the repair machinery in cells? The consensus model of Mfd binding and activation upon engagement with RNAP suggests that engagement of Mfd's RID with the β-subunit of RNAP releases the auto-inhibition of Mfd, allowing the protein to expose its N-terminal BHM that subsequently engages UvrA(reviewed in ref. [54]). Available data do not suggest the presence of additional regulatory mechanisms that might result in lesion-dependent engagement of UvrA once the BHM is exposed. Nevertheless, subtle differences in the architecture of the various Mfd-RNAP complexes and their engagement with UvrA cannot be ruled out in the absence of structural data. As explained above, the tight coupling between Mfd and UvrB detected in cells can be readily interpreted as the initial stages of TCR at sites of endogenous lesions that stall RNAP. In this case, directional loading of UvrB would be predicted to lead to the formation of a pre-incision complex in which UvrB is long lived. On the other hand, at class II pause sites and sites of nucleoprotein roadblocks, one might expect that $UvrA_2B$ are recruited in vivo resulting in unstable loading of UvrB in the absence of DNA damage. Finally, does the recently described release and catch-up mechanism[12] also recruit UvrAB to transcribed genes, potentially ensuring that subsequent rounds of transcription occur on undamaged templates? Resolution of these questions will require dual color studies of fluorescently tagged UvrAB and Mfd proteins.

Mfd has recently been implicated as an 'evolvability factor' responsible for promoting mutagenesis in a host of bacterial pathogens in response to antibiotic treatment[55]. This activity was shown to arise out of interactions with UvrA and RNAP, but as yet no mechanistic explanation is available. Our findings provide two potential explanations for this activity: First, since Mfd sequesters UvrA on DNA in cells (this work and ref. [32], also

proposed in ref. [55]), UvrAB might be unavailable for error-free repair of bulky lesions on DNA. Second, if Mfd recruits UvrAB to sites of paused TECs, then loading of UvrB may promote gratuitous repair[56] at these sites (discussed in ref. [57]). Finally, a third potential explanation may be found in considering the outcomes of the interactions of Mfd with paused RNAP. It has been suggested that under certain conditions Mfd promotes displacement of RNAP at pause sites, leaving behind an R-loop that promotes genomic instability by a host of other mechanisms[58], however, it should be noted that the details of the mechanism, involvement of NER factors, and relevance to Mfd mediated mutagenesis remain to be investigated. How the interactions characterized here can simultaneously give rise to the two contrasting outcomes of error-free repair, and mutagenesis remains to be seen.

## Methods

**Construction of strains and plasmid**. All strains used in this study were derivatives of *Escherichia coli* K-12 MG1655 *mfd-YPet*[21] (see Supplementary Table 1). Point mutations in chromosomal *uvrA* or *uvrB* alleles were introduced by CRISPR-Cas9 assisted λ Red recombination[59,60] (see Supplementary Methods for details and Supplementary Table 2 for sequences of oligonucleotides used to generate mutants). Chromosomal knock-out of *uvrB* was generated previously (ref. [32]) and transduced to other backgrounds as necessary using standard P1 transduction protocols.

pUvrB was created by sub-cloning a geneblock containing the *uvrB* promoter and *uvrB* gene (IDT, Illinois, US; sequence presented in Supplementary Table 3) into pJM1071 (a gift from Woodgate lab)[45] between the *Nde*I and *Xho*I sites. The promoter sequence was identified as 130 nucleotides directly upstream of the *uvrB* gene in the *E. coli* chromosome[25]. Constructs were sequenced on both strands prior to use.

**Cell culture for imaging**. Cells were imaged in quartz-top flow cells as described previously[21,32]. Briefly, flow cells were assembled using a clean quartz piece (Proscitech, Australia) and a bottom cleaned and (3-Aminopropyl)triethoxysilane (Alfa Aesar, A10668, UK)-treated cover-slip (Marienfeld, Deckglaser, 24 mm × 50 mm, No 1.5, German) using double-sided sticky tape (970XL ½ × 36 yd, 3 M, United States), and sealed with 5-min epoxy (Parfix). Quartz top pieces were designed to be able to insert inlet and outlet tubing (PE-60, Instech labs). Prior to imaging, cells were revived from a −80 °C DMSO stock in 500 μL of EZ-rich defined media (Teknova, CA, US), supplemented with 0.2% (v/v) glucose in 2-mL microcentrifuge tubes at 30 °C. Cultures were set to shake in an Eppendorf Thermomixer C (Eppendorf, Australia) at 1000 rpm. On the following day, cultures were reset by inoculating fresh growth medium 1:200 fold, and continued to shake for ~3 h at 30 °C prior to imaging. Cells expressing plasmid-expressed UvrB were grown in growth medium supplemented with spectinomycin (50 μg per mL) to ensure retention of the plasmid. Cells in early exponential phase were loaded in flow cells at 30 °C, followed by a constant supply of aerated EZ-rich defined media at a rate of 30 μL per min, using a syringe pump (Adelab Scientific, Australia).

**Single-molecule live-cell imaging**. Single-molecule fluorescence imaging was carried out with a custom-built microscope as previously described[21]. Briefly, the microscope comprised a Nikon Eclipse Ti body, a 1.49 NA 100x objective, a 514-nm Sapphire LP laser (Coherent) operating at a power density of 71 W cm$^{-2}$, an ET535/30 m emission filter (Chroma) and a 512 × 512 pixel$^2$ EM-CCD camera (Andor iXon 897). The microscope operated in near-TIRF illumination[61] and was controlled using NIS-Elements (Nikon).

Fluorescence images were acquired in time-series format with 0.1-s frames. Each video acquisition contained two phases. The first phase aimed to lower background signal by continuous illumination, causing most of the fluorophores to photo-bleach or to assume a dark state. The second phase (single-molecule phase) is when reactivated single molecules can be reliably tracked on a low background signal. In the second phase, consecutive frames were acquired continuously or interspersed with a dark time ($\tau_d$) as described below.

**Image analysis**. Image analysis was performed in Fiji[62], using the Single Molecule Biophysics plugins (available at https://github.com/SingleMolecule/smb-plugins), and MATLAB. First, raw data were converted to TIF format, followed by background correction and image flattening as previously described[21]. Next, foci were detected in the reactivation phase by applying a discoidal average filter (inner radius of one pixel, outer radius of three pixels), then selecting pixels above the intensity threshold. Foci detected within 3-pixel radius (318 nm) in consecutive frames were considered to belong to the same binding event.

**Interval imaging for dissociation kinetics measurements**. Interval imaging was performed as described previously[21,24]. Briefly, the photobleaching phase (phase I)

contained 50 continuous 0.1-s frames. In phase II, 100 0.1-s frames were collected either continuously or with a delay time ($\tau_d$) ranging from $\tau_d$ (s) = (0.1, 0.2, 0.3, 0.5, 0.9, 1.9, 2.9, 4.9, 7.9, 9.9). In each experiment, videos with varying $\tau_d$ were acquired. Foci were detected using a relative intensity threshold of 8 above the background. Depending on the construct being imaged, between 5–11 repeats of each experiment were collected for each strain. Cumulative residence time distribution (CRTDs) of binding events detected in all data sets were generated for each interval. Lifetimes of DNA-bound Mfd-YPet were determined by globally fitting bootstrapped CRTDs across all intervals using least-squares trust-region reflective algorithms as described in ref. [24]. Minimization was terminated when a tolerance of $10^{-6}$ was reached. Dissociation rates and the corresponding lifetimes are reported as the mean and standard deviations of the bootstrap distribution obtained by repeating the bootstrapping ten times. In each bootstrapping instance, 80% of the raw data set was randomly selected and fit to the models as appropriate to generate the elements of the bootstrap distribution (Supplementary Table 4). See also ref. [24].

**Reporting summary**. Further information on research design is available in the Nature Research Reporting Summary linked to this article.

## Data availability
Data supporting the findings of this manuscript are available from the corresponding author upon reasonable request. A reporting summary for this Article is available as a Supplementary Information file. All source data underlying Figs. 2e, f, 3b–f, 4d–g, and Supplementary Figs. 1–4 are provided as a Source Data file.

## Code availability
Custom code used for data analyses has been made publicly available at github.com/hghodke/bacterial_live_cell_interval_imaging Image analysis was performed in Fiji[62], using the Single Molecule Biophysics plugins (available at https://github.com/SingleMolecule/smb-plugins

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

## Acknowledgements
We thank Lidia G. Alvarez for collecting microscope data during the early stages of the project. H.G. acknowledges the University of Wollongong and the Faculty of Science, Medicine and Health for funding of the Andor iXon 897 used in this work. A.M.v.O. would like to acknowledge support by the Australian Research Council (DP180100858 and FL140100027).

## Author contributions
Construct creation: H.N.H.; Data curation: H.N.H; Data analysis: H.N.H. and H.G.; Software: H.N.H. and H.G.; Writing—Original draft: H.N.H. and H.G.; Writing—review and editing: H.N.H., H.G. and A.M.v.O.; Conceptualization: H.G.; Supervision: H.G. and A.M.v.O.

## Competing interests
The authors declare no competing interests.
