## [Peer Review File · Nature Communications]

Reviewers' comments:

Reviewer #1 (Remarks to the Author):

Understanding how inhibition of RNA polymerase (RNAP) at a site of damage is coupled to the subsequent repair of the lesion is a fundamental question in the field of DNA repair. This outstanding team has recently developed a robust imaging approach to follow single molecules of the bacterial transcription-coupling factor, Mfd, at sites of stalled RNAP (Nature Comm. 2018). In this first paper they showed that deletion of the nucleotide excision repair (NER) protein, UvrA, causes increased dwell time of Mfd on DNA presumably bound to RNAP. In this new study, the authors further study how the kinetics of Mfd dissociation from DNA, presumably at stalled RNAP molecules, is affected by loss of UvrB or specific mutants of UvrA or UvrB. What they find is that UvrB is the key NER protein, which allows Mfd to dissociate from DNA. The experiments are well described and executed, and the work represented a new standard in the field. However, some of these excellent data are over interpreted, and without additional confirmatory experiments, the authors should be more cautious in their overall interpretations. It's clear that alterations in UvrA and UvrB have profound effects on the half-life of Mfd bound to DNA. However, whether Mfd is bound to stalled RNAP and whether RNAP remains bound to Mfd during the course of these observations is not entirely clear. The impact of this work would be enhanced if the authors consider the following points:

1. Please give appropriate references for lines 51-53 and also for the two different models shown in Figure 1, in the figure legend.
2. While the authors infer from their data, that UvrB is actively loaded at Mfd sites, a parsimonious explanation is that UvrB is required to help dissociate both UvrA and Mfd presumably from a site of stalled RNAP. However, without direct evidence that UvrB is actually loaded on the DNA at these sites, sentences like those found on lines 65-67, "...clearly demonstrate that the loading of UvrB on DNA completes the handoff providing a mechanism for the strand- and site-specific loading of the repair machinery at the stall site." while consistent with their data but indirect and the author are urged to be more cautious. The same concern is raised for the interpretation given on lines 150-151, and later on line 160. The same is true for line 192, "In this work, we show that the dissociation of Mfd is facilitated by the loading of UvrB in live bacterial cells." While their data is consistent with a model in which UvrB is loaded, nowhere in this study have they shown UvrB is being loaded at sites of Mfd-DNA complexes. These points have to be softened. Also see point #9.
3. The lifetimes of Mfd are based on cumulative residence time distributions which occur over three orders of magnitude. The life time measurement given in Figure 2 represents how many separate CRTD measurements? It would be helpful to add the total n for each experiment for each figure and perhaps in table 1.
4. Lines 101-102, Even though it has been observed in vitro with purified proteins, without a two color experiment, the presence of UvrA in the Mfd-DNA complex is inferred, the word, "indicate" should be softened to "is consistent with" or "suggests". The authors do indicate later on lines 133-138, despite considerable effort making a red version of UvrA to follow its colocalization with Ypet-tagged Mfd, failed.
5. Lines 106-109, it is not clear from the data or the explanation given, how the authors, in the absence of additional information, can reach the conclusions about RNAP dissociation and longer retainment of Mfd-UvrA. Please clarify.
6. Difference between the beta-hairpin deletion versus Y96A mutants of UvrB and their interpretation. Work by Skorvaga et al, (JBC 2002, 2004) showed that neither mutant "can be loaded on DNA". What they showed was that these UvrB mutants can form a complex with UvrA on DNA, but do not form a stable UvrB-DNA complex by EMSA. Also, UvrC incision at the damaged

site by the beta-hairpin deletion versus Y96A is reduced to less than 5% of WT. Additional data in these two papers indicated that insertion of the beta-hairpin into the DNA helps destabilize the DNA allowing effective UvrA dissociation and incision. Thus, it might be expected that these two UvrB mutants would cause an increase in the life time of Mfd complexes, which is exactly what they observed. The Y96A UvrB mutant shows a less drastic phenotype in their assay.

7. Line 176, "... UvrA is necessary for UvrB to be localized on DNA". This statement is only partially true, UvrB cannot interact with DNA efficiently, but can also interact with UvrC in the absence of UvrA and interact effectively with DNA (Springall et al, NAR 2018). This point needs to be better clarified.

8. It is not clear how the authors concluded, on lines 178-19, "UvrB does not simply compete off Mfd from UvrA2 complexes at the binding interface occupied by Mfd." Their data clearly show that a functional interaction of UvrB with UvrA greatly decreases the lifetime of Mfd on DNA. The experiment showing that high levels of UvrB decrease the Mfd lifetime would seem to support the idea that UvrB can in fact compete.

9. Please provide the citation for lines 227-229. Its important to note that Savery (PNAS 2012) and coworkers have proposed a model in which Mfd can help load UvrAB at sites of paused RNAP, and these UvrAB complexes can scout ahead for damage on the transcribed strand suggesting an uncoupling of Mfd-UvrAB from the stalled RNAP site. This is inconsistent with what the authors describe on lines 240-241 "...leads to recruitment of UvrAB leading to subsequent arrest of Mfd's translocation"

10. Lines 267-268, the original findings by Selby and Sancar (Science 1993) showing dissociation of RNA with RNAP seems to be inconsistent with this explanation.

11. The authors might want to consider the overall stoichiometry of RNAP, Mfd, UvrA and UvrB under non-SOS conditions in E.coli. If RNAP and Mfd greatly exceeds UvrA and UvrB, then if all RNAP were stalled by Rifamycin, then a significant fraction of Mfd lifetimes should collapse down to 30 sec (regardless of the UvrA or UvrB status), like a UvrA deletion strain.

12. While the authors failed to be able to follow labeled UvrA and Mfd in the same cell, perhaps the definitive experiment would be to show that once labeled UvrB arrives at a site of Mfd on DNA, Mfd dissociates.

Reviewer #2 (Remarks to the Author):

The manuscript by Ho, van Oijen, and Ghodke presents single molecule fluorescence studies of stalled RNA polymerase (RNAP) handoff to nucleotide excision repair (NER). Their measurements were performed in vivo in live E. coli cells using a fluorescent fusion protein variant of the prokaryotic transcription-repair coupling factor Mfd. The authors aim at resolving the intermediate protein complex species involved in as well as the order of events in the initiation of transcription coupled NER (TC-NER) by stalled RNAP. Towards this aim, they exploit a range of E. coli strains to determine DNA residence times of Mfd in different protein contexts. Specifically, they use cells containing different mutant variants of either UvrA or UvrB, the two NER proteins involved in the handoff. These variants were defective in ATPase activities (UvrA) or DNA lesion interactions (UvrB). Comparison of these results with those in cells providing a wild-type background as well as with previously published data from their laboratory leads the authors to determining the roles of protein interactions and catalytic activities in the handoff. They interpret their findings in terms of a revised mechanistic model of the handoff in TC-NER initiation.

The study uses an attractive, relatively novel approach for measurements inside living cells at the

single molecule level. The modification of the protein background in these cells by either completely deleting UvrA or UvrB or introducing mutations in the UvrA or UvrB proteins is an elegant method to directly probe effects of the interrupted functions on the molecular interactions in the stalled RNAP-TCR handoff process. The authors have already recently published similar studies on Mfd interactions with RNAP (and on UvrA interactions in TCR initiation, still in revision). I find their results very interesting and informative in the context of the order of events in the Mfd-UvrA2-UvrB complex.

Overall, the study left me with interesting insights but some confusion. I think the conclusions drawn by the authors are overall correct. In particular, I like their conclusion figure (Figure 5) that summarises their findings in the context of the handoff model. Although the story is rather short, this may be of course ideally suited for a Communications feature. One concern would be with the fact that stalling of RNAP by induction of (NER) lesions is not addressed (only undamaged DNA / endogenous damage here), but seems important in the context of initiation of TC-NER. In its current state I would be reluctant to recommend the manuscript for publication (in a high impact factor journal), but given minor changes in the text (especially explanations of observations/interpretations) and some additional information (e.g. *in vitro* affinities of the involved protein variants) would make it a very interesting piece of work well suited for publication in Nature Communications.

Specific points:

1. It is very laudable that the authors have included a Table that summarises their results. Since they frequently refer to residence times of Mfd as well as other proteins in the cascade determined in previous studies by their laboratory, and use these residence times in the interpretation of their data, it would be highly beneficial to the reader if they could include these previous informations in the Table also (marked accordingly). Especially when the authors directly compare, e.g. the effects of UvrB levels, or when they compare residence times of UvrA from previous work with those of Mfd (lines 216/217).
2. There appear to be no DNA damages induced in these experiments. How would this compare to looking at stalled RNAP events and TCR initiation at NER lesions?
3. Lines 103-106 "UvrA can arrest the translocation of Mfd *in vitro* with a lifetime of 15s, however it should be noted that in these experiments, the investigators observed the loss of RNAP, whereas our measurements reflect the loss of Mfd. To reconcile these observations in the absence of UvrB...". Phrasing seems a little unclear. 1) While it can safely be assumed that this automatically includes Mfd in the complex, the phrasing may be a little unclear. 2) More unclear yet seems whether UvrB was present or not in the reported study (in which RNAP loss was observed, especially since 15 and 18 s residence times of RNAP and Mfd would be so temptingly similar), although from the following text it becomes clearer that it was not.
4. How do the authors reconcile the fact that the only two conditions with a "fast lifetime" species (non-specific binding of Mfd to DNA) are UvrA-K646A deltaB (with defective UvrB-interactions in UvrA and no UvrB), and enhanced UvrB? Was it present in the absence of UvrA?
5. Line 146: it might be made clearer that the authors talk about DNA loading here.
6. In this context also in line 120: if the authors refer to DNA loading here not loading of UvrB on the mutant UvrA protein, then is it known on what this strongly reduced loading is based? Are the affinities between UvrA(K646A) and UvrB known (or UvrA(K36A) and UvrB)? Could the authors measure KD's for this UvrA variant (versus wild type UvrA, and UvrA(K37A)) and Mfd?
7. Why would such different lifetime populations be indistinguishable in single molecule analyses? What do the authors suggest is the reason for different retention times on DNA in the absence of UvrB with wild-type UvrA versus in the absence of UvrB with UvrA defective in loading UvrB onto DNA?
8. In the discussion, lines 209/210: could the authors speculate how they envision this effect of UvrB loading on DNA on Mfd dissociation from the complex?
9. Lines 222-224: I am not sure if this is correct? Was it not 15 s for UvrA alone and 50% (although this does not affect the argument here)? The authors explain the apparent discrepancy between effects of UvrA in the absence of UvrB in previous *in vitro* and their own experiments

nicely with the fact that different protein components (RNAP in the in vitro, Mfd here) were detected in the two cases and incorporate the information that results from this (faster dissociation for RNAP than for Mfd-UvrA2) into their summarizing Figure 5. They should mention this Figure here in the text to ease understanding of their conclusions. I am in fact not sure that Figure 5 is currently mentioned at all in the text.

Reviewer #3 (Remarks to the Author):

This manuscript by Ho et al. monitors the dissociation of fluorescently-tagged Mfd from static foci in living *E. coli* using an interval-imaging approach to deconvolve dissociation of Mfd vs. photobleaching of its tag. Using this approach and a series of strains with mutations in UvrA or UvrB, the authors attempt to reconstruct the mechanistic series of events between UvrB, UvrA and Mfd which lead to removal of fluorescent Mfd from DNA.

Although the approach is of great interest to the field and the series of papers from Ho et al. are a pleasure to read and think about, there are a number of misrepresentations and shortcuts which must be corrected for the work to provide clarity, rather than additional confusion, to the field. One such misrepresentation lies in the fact that the authors do not in fact study the system in the context of DNA damage and repair, as the bacteria are never exposed to relevant DNA damaging agents in this manuscript. Any use of the terms "transcription-coupled repair" ("TCR") in this context is therefore misleading and must be scrubbed. Instead, the authors are studying a process more akin to transcription reactivation rather than transcription-coupled repair. Perhaps "release-and-catch-up" as per the work of Le et al. (*Cell*, 2018). Because the complexes studied in this manuscript are not engaged in TCR, it is difficult to draw conclusions about how handoff between components truly engaged in TCR would occur, or what its rules, requirements for partners, or kinetics may be. I return to this at the end of this general commentary.

The consequences of this are important given the unique kinetics of bonafide Mfd-RNAP repair complexes. Indeed, it has been shown that the lifetime of Mfd on DNA depends on the state of the RNAP with which it is interacting. The experiments of Le et al. (*Cell*, 2018) show that when Mfd acts on a paused RNAP, it "bumps" it forward and reactivates the RNAP, but then itself rapidly dissociates from the DNA within times of less than a minute. This is to be compared with the experiments described in Howan et al. (*Nature*, 2012), Graves et al. (*NSMB*, 2015), and Fan et al., (*Nature*, 2016) which conclusively show that when Mfd acts on lesion-stalled RNAP (which can never be reactivated) the resulting Mfd-RNAP complex remains on DNA for thousands of seconds if not longer. The mechanism for early release of Mfd coupled to reactivate-able RNAP remains unknown, but so far no other single-molecule group has made the effort to characterize the full system on bonafide lesions, leading to significant confusion for non-specialists.

To return to the end of the prior paragraph. As the authors have understood, based on published work and experimental conditions used it is this reviewer's opinion that Mfd acts differently on RNAP which is stalled on a lesion vs. RNAP which can be reactivated. Mfd-RNAP formed on a lesion-stalled RNAP is more strongly associated with DNA than Mfd associated alone with DNA after reactivating RNAP (which then runs off on its own). One could think of a "tight" complex in the former case and a "loose" complex in the latter case. In the former "tight" case UvrA action may be sufficient to remove Mfd, whereas in the "loose" case UvrA action may be futile as it attempts to act on a piece of Mfd which, lacking RNAP, is not in fact properly positioned for UvrA-alone mediated removal. In the "tight" case the additional presence of UvrB enhances UvrA catalysis and increases the rate of dissociation of Mfd-RNAP. In the "loose" case it is only in the additional presence of UvrB that UvrA rate or conformational changes finally succeed at removing the "loosely" associated Mfd.

Additional Comments

-It would have been very nice to have the lifetime of Mfd-YPet in UvrA-/UvrB- conditions for the sake of completeness, but I understand that may not be possible given realistic working situations of people.

-Lines 19, 26, 28, 33, 38, 42, 57, etc. : As discussed above there is no transcription-coupled repair occurring here as there is no damage being generated in the DNA, and so a term other than the clearly-defined "TCR" should be found. Reading the paper as "release-and-catch-up" would work and keep things neat in people's minds.

-Line 31: NER can also repair abasic sites.

-Line 33: As earlier...has Mfd interacting with paused RNAP been shown to lead to TCR? Fan et al 2016 and Park et al 2002 are cited but they do not show Mfd interacting with paused RNAP, only stalled RNAP.

-Line 73: As earlier...what is the interpretation of the 29s lifetime (in UvrA- background)? In vitro experiments show without UvrA the Mfd-RNAP intermediate complex is long-lived (at least 100s of seconds)

-Line 104: I recommend rephrasing the sentence to read "UvrA can arrest the translocation of Mfd in vitro for a time of approximately 15s"

-Line 108: The proposal that UvrA can remove RNAP from Mfd-RNAP, but remain stable with Mfd on DNA, lacks evidence and is in disagreement with the fluorescence assays performed by the Strick lab (see below).

-Line 148: "To reconcile our observations with these findings, we propose that the apparently single population detected in our experiments represents a mixture of two populations with indistinguishable lifetimes—one that successfully loads UvrB like wild-type UvrA does, and a population that is unable to load UvrB efficiently." This appears contradictory with the overall manuscript; if there are two separate populations that arise via successful loading or unsuccessful loading of UvrB by UvrA(K37A), how do they have the same lifetimes? The paper specifies that UvrA(K37A) alone is unable to release Mfd rapidly (Mfd-YPet on DNA for 304 s in these conditions), so shouldn't this be the timescale for Mfd-YPet dissociation in the case of "unsuccessful loading" – and quite different from the observed 52 seconds?

-Line 171: There may be a typo here as the data seem to say the opposite? Indeed there appears to be no significant difference between the lifetime of Mfd-YPet when UvrA is present but UvrB is absent (143 +/- 18 s) or when UvrA is present but UvrB is mutated in its beta-hairpin (188 +/- 46 s). The logical conclusion is therefore that UvrB mutated in its beta-hairpin cannot in fact promote disassembly of the Mfd-UvrA2-UvrB complex. Perhaps the authors inverted the beta-hairpin and Y96A mutants in the text? This would explain the statement regarding a four-fold rate different wrt wild-type, which was measured at 18 s.

-Line 212: why is UvrA (or more precisely, UvrA-Mfd-RNAP) not indefinitely arrested in cells lacking UvrB? If I am not mistaken, it can be seen to reside on DNA for 97s; yet Mfd-YPet in these same conditions releases in 143 seconds. As the difference in the two numbers begins to cross the minimal threshold for statistical significance, could this be clarified?

-Line 212: A question arose here regarding this aspect of the model. Because the Mfd-RNAP complex translocates along DNA after RNAP has been removed from damaged DNA, there is no guarantee that it is near a lesion when UvrA or UvrAB find it. Thus if UvrA-Mfd dissociate together,

and because UvrA is required for loading of UvrB onto a lesion, how can downstream coupling take place if UvrA has already left the DNA?

-Line 217: Please clarify why simultaneous dissociation implies strand-specific UvrB loading.

-Line 223: I returned to the paper under discussion; the release time cited was 15 +/- 3 s, not less than 10s.

-Line 230-236: This section is problematic in its representation of Fan et al.: fluorescence experiments presented in Fan et al. showed that both Mfd and RNAP are released by UvrAB, essentially in a simultaneous fashion. The authors should correct their statement and model.

-Line 237: Again, this model does not account for the continued translocation of Mfd-RNAP after remodeling of lesion-stalled RNAP (see query regarding line 212, above).

-Line 255: Similarly to line 33, there is also the 'release and catch-up' mechanism for Mfd interacting with paused RNAP, and this has not been shown to be involved in TCR. This should be mentioned.

-Line 267 : Based on this model, R-loops form every time Mfd removes RNAP. Again, there should be mention of the release and catch up model, and it made clear that currently there is no evidence TCR results from Mfd interaction with paused RNAP

-Figure 5: should be corrected: currently we only know that Mfd interacts with stalled RNAP, not paused RNAP. Sequential UvrA B loading is shown but not mentioned in the text. Order of events in figure legend should be made more clear.

Response to reviewers:

We thank the reviewers for their constructive comments aimed at enhancing the work. We have now clarified the text as follows:

Abstract line 16: rephrased 'catalytic requirements for faithful completion of the handoff during transcription-coupled repair' to 'catalytic requirements for faithful recruitment of nucleotide excision repair proteins'

Abstract line 19: rephrased 'TCR initiation' with 'interactions of Mfd with its partner proteins'

Line 31: replaced 'the presence of bulky, helix-distorting lesions' with 'the presence of lesions' (point raised by reviewer 3)

Line 33: To summarize the function of Mfd more precisely, 'stalled or paused RNAP' is replaced with 'stalled transcriptional complexes at sites of lesions, pause sites or roadblocks' (point raised by reviewer 3)

Line 34-45: added a summary of the literature on the structural properties of Mfd to provide context

Line 66-67: added references (point 1 by reviewer 1)

Line 68-75: added the 'release and catch-up' mechanism (point raised by reviewer 3)

Line 76-102: added context/motivation for the current work

Line 103-104: for clarity, added 'questions on the nature of interactions of Mfd with RNAP and the UvrAB proteins'

Line 109 – 110: added references to conditions under which Mfd participation in TCR (and hence Mfd-UvrAB interactions) should be detected as argued by experts in the field.

Line 114: rephrased 'clearly demonstrate' to read 'findings lead us to propose a model in which', per suggestions by reviewer 1

Line 150: replaced 'indicate' with 'suggest', per suggestion by reviewer 1

Line 153: moved the section describing our attempts at making RFP fusions up.

Line 164: replaced 'TCR handoff' with 'Mfd-UvrA₂'

Line 189: added 'on DNA' for clarity (point 5 by reviewer 2)

Line 192: rephrased 'indistinguishable' with 'unresolvable' for clarity (point raised by reviewer 3)

Line 193: rephrased 'successfully loads UvrB like wild-type UvrA does' with 'can potentially load UvrB on DNA', per suggestions by reviewer 1

Line 214-222: added explanation for the difference between Mfd-YPet lifetimes in *uvrB*(Y96A) and *uvrB*(Δ BHG), in response comments raised by reviewer 1 (point 6) and 3

Line 225: added 'Since mutant UvrB can arrest Mfd on DNA, we infer that' for clarity (point 8 by reviewer 1)

Line 240-242: rephrased 'we show that the dissociation of Mfd is facilitated by the loading of UvrB in live bacterial cells' with 'we demonstrate that in live bacterial cells dissociation of Mfd requires ATP hydrolysis at the proximal site of UvrA, and a UvrB protein with an intact β -hairpin motif', per suggestions by reviewer 1

Line 246-298: restructured the discussion to put our findings in context of other's works (point 9 raised by reviewer 1, and general comments by reviewer 2 and 3)

Line 308: added 'on DNA' for clarity

Line 309 - 317: Clarified why our data rule out a model for competitive binding of UvrB to the Mfd binding site on UvrA (in response to reviewer 1).

Line 323-346: we discuss the implication of UvrB-facilitated dissociation of Mfd in relation to strand-specific loading of UvrB (points raised by reviewer 2 and 3)

Line 347 – 387: Restructured the text to discuss the implications of the coupling between the transcription and repair machineries detected in this work and the potential presence of lesions, pause sites or roadblocks. This discussion is provided to highlight the point that the interactions detected here appear to be blind to the cause of failure of the elongation complex. Potentially our findings describe not just transcription coupled repair but also transcription termination at sites of roadblocks and transcription reactivation described by Le et al. Cell. We also point out that these ideas await further testing. This discussion is in response to R#3 comments.

Line 395 – 399: Elaborated on this discussion in response to R#1 and R#3.

Table 1: added the number of repeats for each experimental condition, per suggestion by reviewer 1. Also, added values from published work (Ho et al., 2018) for the purpose of comparison, per suggestion by reviewer 2 (point 1).

Point by point response to the reviewers' concerns are presented in red text below.

Reviewers' comments:

Reviewer #1 (Remarks to the Author):

Understanding how inhibition of RNA polymerase (RNAP) at a site of damage is coupled to the subsequent repair of the lesion is a fundamental question in the field of DNA repair. This outstanding team has recently developed a robust imaging approach to follow single molecules of the bacterial transcription-coupling factor, Mfd, at sites of stalled RNAP (Nature Comm. 2018). In this first paper they showed that deletion of the nucleotide excision repair (NER) protein, UvrA, causes increased dwell time of Mfd on DNA presumably bound to RNAP. In this new study, the authors further study how the kinetics of Mfd dissociation from DNA, presumably at stalled RNAP molecules, is affected by loss of UvrB or specific mutants of UvrA or UvrB. What they find is that UvrB is the key NER protein, which allows Mfd to dissociate from DNA. The experiments are well described and executed, and the work represented a new standard in the field. However, some of these excellent data are over interpreted, and without additional confirmatory experiments, the authors should be more cautious in their overall interpretations. It's clear that alterations in UvrA and UvrB have profound effects on the half-life of Mfd bound to DNA. However, whether Mfd is bound to stalled RNAP and whether RNAP remains bound to Mfd during the course of these observations is not entirely clear. The impact of this work would be enhanced if the authors consider the following points:

1. Please give appropriate references for lines 51-53 and also for the two different models shown in Figure 1, in the figure legend.

We thank the reviewer for pointing this out. We have now included references.

2. While the authors infer from their data, that UvrB is actively loaded at Mfd sites, a parsimonious explanation is that UvrB is required to help dissociate both UvrA and Mfd presumably from a site of stalled RNAP. However, without direct evidence that UvrB is actually loaded on the DNA at these sites, sentences like those found on lines 65-67, "...clearly demonstrate that the loading of UvrB on DNA completes the handoff providing a mechanism for the strand- and site-specific loading of the repair machinery at the stall site." while consistent with their data but indirect and the authors are urged to be more cautious. The same concern is raised for the interpretation given on lines 150-151, and later on line 160. The same is true for line 192, "In this work, we show that the dissociation of Mfd is facilitated by the loading of UvrB in live bacterial cells." While their data is consistent with a model in which UvrB is loaded, nowhere in this study have they shown UvrB is being loaded at sites of Mfd-DNA complexes. These points have to be softened. Also see point #9.

We agree with this reviewer that we have not provided evidence that UvrB is indeed loaded at these sites. We were limited in this endeavor due to our inability to create UvrB fusions to red fluorescent proteins that would have permitted us to gather evidence from dual color experiments. For this reason, we were left with no other option but to infer that UvrB is loaded based on the use of beta-hairpin mutants of UvrB that have previously been demonstrated to be deficient in engaging DNA in EMSAs.

We have now softened the language to indicate that the potential for stable engagement with DNA is a key attribute required for completion of the handoff.

Line 114: we have replaced "clearly demonstrate" with "findings lead us to propose a model in which"

Line 193: "successfully loads" is replaced with "can potentially load UvrB"

Line 240-242 is now rephrased to read: "we demonstrate that in live bacterial cells dissociation of Mfd requires ATP hydrolysis at the proximal site of UvrA, and a UvrB protein with an intact β -hairpin motif."

3. The lifetimes of Mfd are based on cumulative residence time distributions which occur over three orders of magnitude. The life time measurement given in Figure 2 represents how many separate CRTD measurements? It would be helpful to add the total n for each experiment for each figure and perhaps in table 1.

We thank this reviewer for this suggestion. The CRTD in figure 2 is constructed from 8 independent repeats and the number of counts in each distribution ranges from 2620 – 4900. These statistics are additionally available in the source data file where the raw data have been included. We have now additionally included these in the figure legends and added a column in table 1.

4. Lines 101-102, Even though it has been observed in vitro with purified proteins, without a two color experiment, the presence of UvrA in the Mfd-DNA complex is inferred, the word, "indicate" should be softened to "is consistent with" or "suggests". The authors do indicate later on lines 133-138, despite considerable effort making a red version of UvrA to follow its colocalization with Ypet-tagged Mfd, failed.

We agree and we have now changed indicate to 'suggest' in line with this reviewer's suggestion (line 150).

5. Lines 106-109, it is not clear from the data or the explanation given, how the authors, in the absence of additional information, can reach the conclusions about RNAP dissociation and longer retainment of Mfd-UvrA. Please clarify.

Our work does not allow us to comment on the residence time of RNAP after engagement with Mfd. Lines 106-109 merely represented an attempt to unify and reconcile our observations with previous single-molecule data from the Strick lab. We have now removed this discussion entirely in the interest of clarity.

6. Difference between the beta-hairpin deletion versus Y96A mutants of UvrB and their interpretation. Work by Skorvaga et al, (JBC 2002, 2004) showed that neither mutant “can be loaded on DNA”. What they showed was that these UvrB mutants can form a complex with UvrA on DNA, but do not form a stable UvrB-DNA complex by EMSA. Also, UvrC incision at the damaged site by the beta-hairpin deletion versus Y96A is reduced to less than 5% of WT. Additional data in these two papers indicated that insertion of the beta-hairpin into the DNA helps destabilize the DNA allowing effective UvrA dissociation and incision. Thus, it might be expected that these two UvrB mutants would cause an increase in the life time of Mfd complexes, which is exactly what they observed. The Y96A UvrB mutant shows a less drastic phenotype in their assay.

Indeed, these are the very reasons why we suggest that UvrB loading is critical for Mfd dissociation. Further, in our experiments with UvrA, we have similarly seen that beta hairpin mutants of UvrB prolong the residence time of UvrA on DNA (see Ghodke et al. 2019 bioRxiv, <https://doi.org/10.1101/515502>).

Regarding the stability of UvrA-UvrB(hairpin mutants) on DNA using EMSAs – this is quite tricky to interpret for two reasons:

1. Kad et al 2010 demonstrate that UvrAB can freely diffuse on DNA under certain conditions. We are not surprised that UvrB that cannot stably engage DNA, might fall off the short DNA templates that are typically used in EMSA studies.
2. In our experiments, presumably the Mfd serves to anchor the UvrA₂-UvrB(mutant) proteins to the site where it is recruited permitting us to measure the lifetimes reliably.

For these reasons, we believe that the long lifetimes measured here are not inconsistent with the EMSA experiments performed with UvrA₂-UvrB(hairpin mutants)₂. The appropriate experiment for purposes of comparison would be Mfd-UvrA₂-UvrB(mutant) complex lifetimes measured *in vitro* – however, to our knowledge, this experiment remains to be performed.

We have elaborated on this point in lines 214 – 222.

7. Line 176, “... UvrA is necessary for UvrB to be localized on DNA”. This statement is only partially true, UvrB cannot interact with DNA efficiently, but can also interact with UvrC in the absence of UvrA and interact effectively with DNA (Springall et al, NAR 2018). This point needs to be better clarified.

Indeed. This reviewer is correct. We apologize for the confusion. Our statement was made strictly in the context of recruitment of UvrB to sites of Mfd. We have now rephrased this to clarify this point.

8. It is not clear how the authors concluded, on lines 178-19, “UvrB does not simply compete off Mfd from UvrA₂ complexes at the binding interface occupied by Mfd.” Their data clearly show that a functional interaction of UvrB with UvrA greatly decreases the lifetime of Mfd on DNA. The experiment showing that high levels of UvrB decrease the Mfd lifetime would seem to support the idea that UvrB can in fact compete.

There are two possible ways in which UvrB may influence the lifetime of Mfd.

Model 1. UvrB could replace Mfd bound to UvrA₂(B) so that the Mfd-UvrA₂(-UvrB) complex yields UvrB(2)-UvrA₂

Model 2. UvrB could bind UvrA to form the Mfd-UvrA-UvrA-UvrB complex which is then resolved by ATPase activity of UvrA leading to loss of Mfd, and UvrA₂ simultaneously leaving behind UvrB at the site.

We demonstrate that:

- a. over expression of UvrB truncates the lifetime of Mfd (consistent with model 1 and 2)
 - b. beta-hairpin mutants of UvrB prolong the lifetime of Mfd (only consistent with model 2).
- These mutants are unaffected in their ability to interact with UvrA (Skorvaga JBC, 2002).

Of the two scenarios described above, only the second explanation accommodates both experimental findings. We propose that the lifetime of Mfd on DNA is composed of (at least two components) – first, time taken to execute catalysis and second, time spent in waiting for UvrB. Additionally, overexpression of UvrB would raise the concentration in the cell, which would lead to shorter search times, consistent with our observation that the lifetime is truncated when UvrB is expressed at a higher concentration. This is now clarified in 309 - 317.

9. Please provide the citation for lines 227-229. Its important to note that Savery (PNAS 2012) and coworkers have proposed a model in which Mfd can help load UvrAB at sites of paused RNAP, and these UvrAB complexes can scout ahead for damage on the transcribed strand suggesting an uncoupling of Mfd-UvrAB from the stalled RNAP site. This is inconsistent with what the authors describe on lines 240-241 "...leads to recruitment of UvrAB leading to subsequent arrest of Mfd's translocation"

We apologize for the confusion arising from a typo. 'UvrA₂B' should have read 'UvrA₂'. We have now clarified this point in the text.

After either reactivating RNAP or disassembling it, Mfd may continue to translocate on DNA (as shown by Strick lab). The wait time before UvrA₂ arrives at the scene will determine the extent of Mfd's translocation (4 nt/s Strick lab estimate). The model built on our findings completely accommodates both Mfd translocation and the seminal findings by Savery lab that Mfd can load UvrAB at downstream sites.

We have now expanded this in the restructured discussion in the text. Changes are highlighted lines 246-298.

10. Lines 267-268, the original findings by Selby and Sancar (Science 1993) showing dissociation of RNA with RNAP seems to be inconsistent with this explanation.

While this is true, Wimberly et al 2013 (PMID: 23828459) suggest a specific scenario where long RNA's may form an R-loop upstream of the RNAP, and the action of Mfd on these stalled RNAPs may lead to retention of the R-loop. In the event that R-loops are indeed left behind, these could be mutagenic. We await further elucidation of this pathway before we can comment on its relevance to the mutagenicity associated with Mfd.

11. The authors might want to consider the overall stoichiometry of RNAP, Mfd, UvrA and UvrB under non-SOS conditions in E.coli. If RNAP and Mfd greatly exceeds UvrA and UvrB, then if all RNAP were stalled by Rifamycin, then a significant fraction of Mfd lifetimes should collapse down to 30 sec (regardless of the UvrA or UvrB status), like a UvrA deletion strain.

We agree that when $[Mfd] \gg [UvrA]$, cells will experience a condition where UvrA is limiting. In that case we would anticipate dissociation rates similar to those in $\Delta uvrA$ cells.

In our strains, the copy numbers per cell are: UvrA-YPet (16; Ghodke et al. 2019 bioRxiv, <https://doi.org/10.1101/515502>), Mfd-YPet (22; Ho et al., NComms 2018) and UvrB-YPet (~24; unpublished). We expect RNAP to be 3000-4000 based on Kapanidis lab estimates. In these strains, $[Mfd]$ is comparable to that of $[UvrA]$ and $[UvrB]$.

We note that NER factor copy numbers:

1. are different from the widely cited numbers from the Sancar lab that are presumably based on antibody staining
2. Appear to be strain dependent – our estimates are different from those presented in Stracy et al 2016 and those in Crowley and Hanawalt 1998.

Finally, rifampicin does not stall RNAP, but prevents the formation of elongating transcriptional complexes. We have reported the rif experiment in Ho et al., Nature Communications, 2018 and found very few binding events of Mfd-YPet.

12. While the authors failed to be able to follow labeled UvrA and Mfd in the same cell, perhaps the definitive experiment would be to show that once labeled UvrB arrives at a site of Mfd on DNA, Mfd dissociates.

Absolutely. We have attempted making mKate2 and PAmCherry1 fusions of UvrA, Mfd and UvrB. IN all three cases, the copy numbers were inconsistent with those observed in our YPet strains. The way forward is to survey several available red fluorescent proteins to identify RFP fusions that are suitable for co-localization followed by lambda red recombination to create strains that are good enough for these complex experiments. Unfortunately, this is beyond the scope of this current study.

Reviewer #2 (Remarks to the Author):

The manuscript by Ho, van Oijen, and Ghodke presents single molecule fluorescence studies of stalled RNA polymerase (RNAP) handoff to nucleotide excision repair (NER). Their measurements were performed *in vivo* in live *E. coli* cells using a fluorescent fusion protein variant of the prokaryotic transcription-repair coupling factor Mfd. The authors aim at resolving the intermediate protein complex species involved in as well as the order of events in the initiation of transcription coupled NER (TC-NER) by stalled RNAP. Towards this aim, they exploit a range of *E. coli* strains to determine DNA residence times of Mfd in different protein contexts. Specifically, they use cells containing different mutant variants of either UvrA or UvrB, the two NER proteins involved in the handoff. These variants were defective in ATPase activities (UvrA) or DNA lesion interactions (UvrB). Comparison of these results with those in cells providing a wild-type background as well as with previously published data from their laboratory leads the authors to determining the roles of protein interactions and catalytic activities in the handoff. They interpret their findings in terms of a revised mechanistic model of the handoff in TC-NER initiation.

The study uses an attractive, relatively novel approach for measurements inside living cells at the single molecule level. The modification of the protein background in these cells by either completely deleting UvrA or UvrB or introducing mutations in the UvrA or UvrB proteins is an elegant method to directly probe effects of the interrupted functions on the molecular interactions in the stalled RNAP-TCR handoff process. The authors have already recently published similar studies on Mfd interactions with RNAP (and on UvrA interactions in TCR initiation, still in revision). I find their results very interesting and informative in the context of the order of events in the Mfd-UvrA2-UvrB complex.

Overall, the study left me with interesting insights but some confusion. I think the conclusions drawn by the authors are overall correct. In particular, I like their conclusion figure (Figure 5) that summarises their findings in the context of the handoff model. Although the story is rather short, this may be of course ideally suited for a Communications feature. One concern would be with the fact that stalling of RNAP by induction of (NER) lesions is not addressed (only undamaged DNA / endogenous damage here), but seems important in the context of initiation of TC-NER. In its current state I would be reluctant to recommend the manuscript for publication (in a high impact factor journal), but given minor changes in the text (especially explanations of observations/interpretations) and some additional information (e.g. *in vitro* affinities of the involved protein variants) would make it a very interesting piece of work well suited for publication in Nature Communications.

We thank the reviewer for their constructive feedback and for appreciating the value of our single-molecule dissection of interactions between protein factors functioning in TCR. We have now extensively restructured the discussion to draw parallels between the various sub-complexes that have been detected in *in vitro* work, and the comparable live cell results from our study. Where possible, we have attempted to provide binding affinities (see response in specific comments below).

Specific points:

1. It is very laudable that the authors have included a Table that summarises their results. Since they frequently refer to residence times of Mfd as well as other proteins in the cascade determined in previous studies by their laboratory, and use these residence times in the interpretation of their data, it would be highly beneficial to the reader if they could include these previous informations in the Table also (marked accordingly). Especially when the authors directly compare, e.g. the effects of UvrB levels, or when they compare residence times of UvrA from previous work with those of Mfd (lines 216/217).

We thank the reviewer for this suggestion. We have now amended table 1 while providing references to the other two papers as appropriate.

2. There appear to be no DNA damages induced in these experiments. How would this compare to looking at stalled RNAP events and TCR initiation at NER lesions?

Indeed, we have not induced exogenous DNA damage in any of the experiments described here. However, we have examined the lifetime of Mfd in response to UV damage in the related m/s (figure 5 in Ghodke et al. 2019 bioRxiv, <https://doi.org/10.1101/515502>). We found that Mfd is turned over faster once SOS is induced, and that the DNA binding lifetimes of Mfd and UvrA are essentially identical. From the perspective of UvrA, interactions with Mfd are prioritized after UV irradiation consistent with prioritization of TCR at sites of lesions. It should be noted that in response to UV irradiation both UvrA and B are rapidly expressed, and the drop in lifetimes of Mfd/UvrA may at least partially be attributable to enhanced copy numbers of UvrAB promoting faster search. We refer the reader to the related m/s (Ghodke et al. 2019 bioRxiv, <https://doi.org/10.1101/515502>) for these data.

Measurement of the lifetime of Mfd at sites of lesion-stalled RNAP is a challenging experiment. We have devised an approach to visualize stalled RNAP at sites of roadblocks, and measure lifetime of Mfd at these sites, however, this work is very preliminary at this stage.

3. Lines 103-106 "UvrA can arrest the translocation of Mfd in vitro with a lifetime of 15s, however it should be noted that in these experiments, the investigators observed the loss of RNAP, whereas our measurements reflect the loss of Mfd. To reconcile these observations in the absence of UvrB...".

Phrasing seems a little unclear.

1) While it can safely be assumed that this automatically includes Mfd in the complex, the phrasing may be a little unclear.

2) More unclear yet seems whether UvrB was present or not in the reported study (in which RNAP loss was observed, especially since 15 and 18 s residence times of RNAP and Mfd would be so temptingly similar), although from the following text it becomes clearer that it was not.

The discussion in lines 103 -109 was an attempt to reconcile our experimental findings with those from the Strick lab. Instead of discussing this twice, we have now removed the

discussion in lines 103-109 this section from the results, and have retained it in the 'Discussion' section (line 269-298).

4. How do the authors reconcile the fact that the only two conditions with a "fast lifetime" species (non-specific binding of Mfd to DNA) are UvrA-K646A deltaB (with defective UvrB-interactions in UvrA and no UvrB), and enhanced UvrB? Was it present in the absence of UvrA?

We have now included a summary of the results for *mfd-ypet duvrA* in table 1. The fast lifetime of Mfd is also present in this case (consistent with K646A). It is curious that in all three cases, fewer binding partners of Mfd are present – either due to deletion of the UvrA gene, or in the form of a UvrA mutant that does not engage Mfd, or when a competitor for the same binding site on UvrA is present in the form of UvrB. One explanation could be that in these backgrounds the proportion of 'free' Mfd available for DNA binding is somewhat higher. We have insufficient experimental evidence to support this hypothesis.

5. Line 146: it might be made clearer that the authors talk about DNA loading here.

We apologize for this confusion. We have now clarified the text (line 189).

6. In this context also in line 120: if the authors refer to DNA loading here not loading of UvrB on the mutant UvrA protein, then is it known on what this strongly reduced loading is based? Are the affinities between UvrA(K646A) and UvrB known (or UvrA(K36A) and UvrB)? Could the authors measure KD's for this UvrA variant (versus wild type UvrA, and UvrA(K37A)) and Mfd?

Orren and Sancar 1989 demonstrate that ATP (but not ATP_gS or ADP) is required for the formation of the UvrA-UvrB complex, suggesting that ATP hydrolysis is the essential for complex formation. Further, loading of UvrB on DNA also requires ATP hydrolysis. In Myles et al, Biochemistry 1991 the investigators use DNaseI footprinting assays to demonstrate that the association of UvrB with UvrA(K646A) and UvrA(K37A) is diminished. Specifically, UvrA(K646A) can load UvrB to a maximum extent of 1% of that of wildtype UvrA on DNA. The UvrA(K37A) mutant retains 10% UvrB loading activity on DNA compared to wildtype UvrA. To our knowledge, binding affinities of UvrB or Mfd to the walker mutants of UvrA have not been published.

Measurement of these binding affinities would require us to create expression constructs and purify each of UvrA, UvrA(K646A), UvrA(K37A), UvrB, Mfd, Mfd(E730Q) or only the BHM, and develop assays for measuring these interactions. We are not currently equipped to pursue this line of *in vitro* investigation. Nevertheless, we agree with this reviewer that these are important measurements that will ultimately support findings in our work (as well as Stracy et al., 2016) and further clarify the mysterious role of UvrA's ATPase activity in damage recognition. However, we submit that this endeavor is beyond the scope of this m/s.

7. Why would such different lifetime populations be indistinguishable in single molecule analyses? What do the authors suggest is the reason for different retention times on DNA in

the absence of UvrB with wild-type UvrA versus in the absence of UvrB with UvrA defective in loading UvrB onto DNA?

We find that when off rates are within the order of magnitude (as appears to be the case here), resolution of off rates becomes tremendously challenging (see Ho et al., *Biophysical journal*, <https://doi.org/10.1016/j.bpj.2019.07.015>). For the case where the two off rates are only within the order of magnitude, our simulations indicate that the ability to resolve the two lifetimes requires greater than 100,000 data points in every cumulative residence time distribution constructed for each dark interval (11 intervals typically). In practice, this would require us to perform more than 100 perfect repeats of each imaging experiment, each lasting 4 hours. Despite such an effort, the ability to resolve the two populations would only be accessible to conditions where the amplitude of the slowly dissociating species is between 0 – 50%. Unfortunately, practical considerations of microscope time make this infeasible.

We propose the differences in Mfd-YPet binding lifetimes in *uvrB*, *uvrA(K646A) uvrB* and *uvrA(K37A) uvrB* are due to:

- Inability of UvrA(K646A) to efficiently engage Mfd, hence Mfd-YPet in this genetic background (*uvrA(K646A) uvrB*) behaves similarly to Mfd in Δ *uvrA* cells, exhibiting a lifetime of 26 s.
- UvrA(K37A) stabilizes Mfd more strongly than wild-type UvrA in *uvrB* cells (304 s vs 139 s). These results lead us to propose that dissociation of Mfd from arrested Mfd-UvrA complex requires ATP hydrolysis at the proximal site.

8. In the discussion, lines 209/210: could the authors speculate how they envision this effect of UvrB loading on DNA on Mfd dissociation from the complex?

We suggest that ATPase induced global conformational changes of UvrA directionally place UvrB in proximity to the DNA strand. Note that current models suggest that UvrB is loaded on the undamaged strand. At this point, we suggest that engagement with the beta hairpin and placement of the undamaged strand in the groove allow UvrB to 'grip' the strand. This gripping, accompanied by conformational changes in UvrA may transduce a strain that allows Mfd to be displaced from the DNA.

We hesitated to originally include this discussion since it is highly speculative, and since structural data describing the intermediates is not yet available. However, in light of these comments, we have now elaborated on this in lines 323 -345 in the revised discussion.

9. Lines 222-224: I am not sure if this is correct? Was it not 15 s for UvrA alone and 50% (although this does not affect the argument here)? The authors explain the apparent discrepancy between effects of UvrA in the absence of UvrB in previous *in vitro* and their own experiments nicely with the fact that different protein components (RNAP in the *in vitro*, Mfd here) were detected in the two cases and incorporate the information that results from this (faster dissociation for RNAP than for Mfd-UvrA₂) into their summarizing Figure 5. They should mention this Figure here in the text to ease understanding of their conclusions. I am in fact not sure that Figure 5 is currently mentioned at all in the text.

We have addressed the comment on the lifetime by considering comments by R#3 – lines 269-279. As R#3 points out, additional experiments do not support our hypothesis. However, we note that neither the rapid loss of beads observed upon addition of UvrA to translocating Mfd-RNAP, nor the relatively short lifetimes of RNAP-Mfd-UvrA₂ complex (Fan et al) are consistent with our findings.

We have now removed this discussion and are at a loss to explain why Mfd-RNAP are seen to dissociate *in vitro*, whereas the situation *in vivo* is quite different. We have now explained this in lines 280 – 298. We note that the single-molecule *in vitro* experiments were not performed at physiological concentrations of the repair proteins which are upwards of 10 nM in cells, and some of the activities detected in the single-molecule *in vitro* experiments may be influenced by the low concentration of downstream factors.

We thank the reviewer for pointing out the lack of citation of Fig. 5 in the previous version of the m/s. Fig. 5 is now referred in line 242. We note that figure 5 represents a unified model for the handoff in the physiological context summarized in a manner that incorporates data from the *in vitro* experiments that are consistent with our observations. We appreciate that RNAP may dissociate quickly from Mfd in cells, however, we are unable to provide evidence for the same currently. We have therefore left this out of the summary figure.

Reviewer #3 (Remarks to the Author):

This manuscript by Ho et al. monitors the dissociation of fluorescently-tagged Mfd from static foci in living *E. coli* using an interval-imaging approach to deconvolve dissociation of Mfd vs. photobleaching of its tag. Using this approach and a series of strains with mutations in UvrA or UvrB, the authors attempt to reconstruct the mechanistic series of events between UvrB, UvrA and Mfd which lead to removal of fluorescent Mfd from DNA.

Although the approach is of great interest to the field and the series of papers from Ho et al. are a pleasure to read and think about, there are a number of misrepresentations and shortcuts which must be corrected for the work to provide clarity, rather than additional confusion, to the field. One such misrepresentation lies in the fact that the authors do not in fact study the system in the context of DNA damage and repair, as the bacteria are never exposed to relevant DNA damaging agents in this manuscript. Any use of the terms "transcription-coupled repair" ("TCR") in this context is therefore misleading and must be scrubbed. Instead, the authors are studying a process more akin to transcription reactivation rather than transcription-coupled repair. Perhaps "release-and-catch-up" as per the work of Le et al. (Cell, 2018). Because the complexes studied in this manuscript are not engaged in TCR, it is difficult to draw conclusions about how handoff between components truly engaged in TCR would occur, or what its rules, requirements for partners, or kinetics may be. I return to this at the end of this general commentary.

The consequences of this are important given the unique kinetics of bonafide Mfd-RNAP repair complexes. Indeed, it has been shown that the lifetime of Mfd on DNA depends on the state of the RNAP with which it is interacting. The experiments of Le et al. (Cell, 2018) show that when Mfd acts on a paused RNAP, it "bumps" it forward and reactivates the RNAP, but then itself rapidly dissociates from the DNA within times of less than a minute. This is to be compared with the experiments described in Howan et al. (Nature, 2012), Graves et al. (NSMB, 2015), and Fan et al., (Nature, 2016) which conclusively show that when Mfd acts on lesion-stalled RNAP (which can never be reactivated) the resulting Mfd-RNAP complex remains on DNA for thousands of seconds if not longer. The mechanism for early release of Mfd coupled to reactivate-able RNAP remains unknown, but so far no other single-molecule group has made the effort to characterize the full system on bonafide lesions, leading to significant confusion for non-specialists.

To return to the end of the prior paragraph. As the authors have understood, based on published work and experimental conditions used it is this reviewer's opinion that Mfd acts differently on RNAP which is stalled on a lesion vs. RNAP which can be reactivated. Mfd-RNAP formed on a lesion-stalled RNAP is more strongly associated with DNA than Mfd associated alone with DNA after reactivating RNAP (which then runs off on its own). One could think of a "tight" complex in the former case and a "loose" complex in the latter case. In the former "tight" case UvrA action may be sufficient to remove Mfd, whereas in the "loose" case UvrA action may be futile as it attempts to act on a piece of Mfd which, lacking RNAP, is not in fact properly positioned for UvrA-alone mediated removal. In the "tight" case the additional presence of UvrB enhances UvrA catalysis and increases the rate of dissociation of Mfd-RNAP.

In the “loose” case it is only in the additional presence of UvrB that UvrA rate or conformational changes finally succeed at removing the “loosely” associated Mfd.

Response to general comments:

We agree with the reviewer that using current methodology it is not possible to formally prove that endogenous DNA damage occurs during normal growth. We introduced changes in our terminology to make it consistent with work from other labs working in the area of TCR. However, a key premise of our work (supported by the others in this field) is that endogenous damage during normal growth does take place and as a result that TCR is a physiologically relevant mechanism even in the absence of damage externally. In agreement with this reviewer’s concern that our use of the term TCR does not adequately describe reality, we have now instead chosen to use the term transcription termination/reactivation to accommodate findings from both the Strick lab and the Roberts/Wang labs.

We respectfully disagree with this reviewer’s position that TCR does not occur in growing cells. Such a position is incongruent with current understanding of TCR functions. In this work (and Ghodke et al. 2019 bioRxiv, <https://doi.org/10.1101/515502>, under review at Nature Communications) we have visualized and quantified the DNA binding properties of fluorescently tagged UvrA, and Mfd in *bona fide* interactions with RNAP (see Ho et al., Nature Communications 2018) in various genetic backgrounds (including wild-type, KO or mutant *uvrA*, *uvrB*) to characterize the network of interactions that have been identified as *bona fide* interactions in transcription coupled repair. The objective of this work was to detect interactions in between Mfd and UvrAB, and to understand how the repair machinery is recruited to DNA in an Mfd dependent manner.

To that end, we chose to perform our experiments in growing cells in the absence of exogenous DNA damage. All living organisms experience DNA damage as a product of metabolic activities, and this damage can impede the progress of RNAP during normal growth. For these reasons, even though quantification of endogenous DNA damage in growing bacterial cells at the single cell level is currently a formidable task (bordering on the infeasible), we felt that normal growth conditions would be a reasonable starting point for our studies.

This opinion is echoed in Portman and Strick (JMB, 2018) who argue that non-SOS induced conditions are appropriate for investigations of TCR. In fact, these authors have argued in Fan ... Strick, Nature, 2016 “The TCR pathway we detail therefore appears to be most relevant to ‘housekeeping’ DNA repair, watchfully maintaining genomic integrity even in the absence of stressful or genotoxic conditions.” Further, in Portman and Strick (JMB 2018), these experts reinforce this argument claiming, “It also underscores one of the important functional roles of TCR: to act as a “background” surveillance mechanism, able to effectively direct the relatively low-concentration NER system to DNA lesions located on the transcribed strand even in the absence of SOS repair and UvrA, UvrB and UvrC induction”.

Whereas it is true that the work from the Strick lab represents the only published single molecule *in vitro* reconstitution of TCR at sites of defined lesions, in our work, we have

dissected the interactions between RNAP, Mfd, UvrA and UvrB using a novel single molecule imaging approach combined with a series of structural and functional mutants of UvrAB. In these experiments, despite being a single molecule approach, we have described measurements from at least 20,000 observations for each condition, often significantly greater.

In fact, we have performed the experiments in exactly the conditions in which detection of TCR activities should be most optimal (as argued by Strick and other key authorities in the field). Our experiments are completely consistent with several key results from the Sancar, Savery and Strick labs (as indicated in the discussion section). Almost all observations in the key papers cited by the referee as describing the mechanism of TCR (Howan et al., Graves et al., and Fan et al; all from the Strick lab) are faithfully recapitulated in our work. Beyond that, these key findings from the Strick lab have been critical for understanding our results.

It is therefore surprising that when presented with experiments that provide an extensive dissection of interactions that are known to occur in TCR, performed under conditions that are ideal for observation of TCR, this reviewer adopts the position that TCR does not occur in growing cells, and any mention of it should be scrubbed from this manuscript.

Instead, the reviewer argues that far from being authentic interactions in TCR, the observations in our experiments must represent the “release and catch-up” mechanism that has been recently described in Le et al., Cell 2017. We note that Le et al, 2017 make no mention of UvrAB in their work, and provide no evidence to suggest that UvrAB are relevant to this transcription reactivation and are recruited to such sites. No mechanistic description of transcription reactivation is currently available.

Acknowledging that the mechanism of transcription reactivation is poorly understood, this reviewer instead proposes a hypothetical model that might explain how Mfd-UvrA interactions may discriminate between RNAPs that are stalled *vs.* those that are merely paused. The problem here is that there is absolutely no experimental support for such a model in the literature (in fact, this model has never been peer reviewed in the literature either) – one would need to provide binding affinities of Mfd for the two types of RNAPs, and further evidence that UvrA engages Mfd bound to the two types of RNAPs differently. Clearly, some more elucidation of the model is necessary on the referee’s part before this model should be considered seriously. Lacking the experimental evidence outlined above, explaining our results based on an unproved, speculative model would violate the scientific method.

Additional Comments

-It would have been very nice to have the lifetime of Mfd-YPet in UvrA-/UvrB- conditions for the sake of completeness, but I understand that may not be possible given realistic working situations of people.

We thank the reviewer for understanding our practical limitations. Since no experimental evidence is available that demonstrates Mfd interactions with UvrB in the absence of UvrA, we did not deem it necessary to make the Mfd-YPet *duvrA duvrB* strain. We expect that the

lifetime of Mfd-YPet in the double knock-out strain would be identical to that in the *uvrA* knock-out but currently do not have the reagents to test this hypothesis.

-Lines 19, 26, 28, 33, 38, 42, 57, etc. : As discussed above there is no transcription-coupled repair occurring here as there is no damage being generated in the DNA, and so a term other than the clearly-defined "TCR" should be found. Reading the paper as "release-and-catch-up" would work and keep things neat in people's minds.

We agree with the reviewer that the term 'TCR' also does not sufficiently capture the scenario that Mfd may reactivate RNAP. We now use the terms 'transcription reactivation/termination' to capture both scenarios when describing our results. However, we respectfully disagree with this reviewer's suggestion that there is no TCR occurring in this case. Mfd is recruited to RNAP that is stalled at lesions or roadblocks or paused at pause sites. We refer this referee to Proshkin and Mironov (2016), Haines et al., 2014 that document Mfd activity and TCR at sites other than lesions characterized in Fan et al. 2016. We cannot dismiss these scenarios as events that do not occur in the cellular environment.

Further, the suggestion to scrub all mentions of TCR in an introduction to the transcription repair coupling factor Mfd, would create great confusion in the minds of readers. We have therefore elected to retain the use of the term TCR in the introduction section as appropriate.

We hesitate to exclusively use the term "release and catch-up" since it does not adequately capture other scenarios in which Mfd may be recruited to RNAP including paused or stalled RNAP at sites of endogenous lesions or protein roadblocks. Further, as this referee is well aware, NER factors are not known to be recruited at sites of transcription reactivation. Additionally it is unclear whether the release and catch up scenario is even relevant inside cells. Hence, we feel that recasting a study that clearly demonstrates the influence of residence time of the NER factors on Mfd inside living cells as a study on transcription reactivation is inappropriate.

-Line 31: NER can also repair abasic sites.

We have now clarified the text (line 31).

-Line 33: As earlier...has Mfd interacting with paused RNAP been shown to lead to TCR? Fan et al 2016 and Park et al 2002 are cited but they do not show Mfd interacting with paused RNAP, only stalled RNAP.

We refer this reviewer to Proshkin and Mironov (2016), Haines et al. (PNAS, 2014) and Le et al. 2017 that additionally describe RNAP configurations that are substrates for Mfd. We have now additionally cited these references (line 33).

-Line 73: As earlier...what is the interpretation of the 29s lifetime (in UvrA- background)? In vitro experiments show without UvrA the Mfd-RNAP intermediate complex is long-lived (at least 100s of seconds)

This has now been extensively discussed in lines 246 – 268 to clarify this issue for readers.

-Line 104: I recommend rephrasing the sentence to read “UvrA can arrest the translocation of Mfd in vitro for a time of approximately 15s”

We have now removed this section however this change is made (see below).

-Line 108: The proposal that UvrA can remove RNAP from Mfd-RNAP, but remain stable with Mfd on DNA, lacks evidence and is in disagreement with the fluorescence assays performed by the Strick lab (see below).

As above we have now removed this section.

-Line 148: “To reconcile our observations with these findings, we propose that the apparently single population detected in our experiments represents a mixture of two populations with indistinguishable lifetimes—one that successfully loads UvrB like wild-type UvrA does, and a population that is unable to load UvrB efficiently.”

This appears contradictory with the overall manuscript; if there are two separate populations that arise via successful loading or unsuccessful loading of UvrB by UvrA(K37A), how do they have the same lifetimes? The paper specifies that UvrA(K37A) alone is unable to release Mfd rapidly (Mfd-YPet on DNA for 304 s in these conditions), so shouldn't this be the timescale for Mfd-YPet dissociation in the case of “unsuccessful loading” – and quite different from the observed 52 seconds?

We apologize for the confusion. We used the term ‘indistinguishable’, where perhaps ‘unresolvable’ might have been a better choice. Changed line 191.

As mentioned in the response to R#2, the resolvability of two exponentially distributed populations with mean lifetimes that are within the order of magnitude is challenging. Based on our simulations (Ho et al., 2019, *Biophysical Journal*, <https://doi.org/10.1016/j.bpj.2019.07.015>), to effectively resolve these, we will require approximately 100,000 counts in every CRTD for each dark interval. In practice this means we need 100 perfect repeats of experiments that last 4 hours each. A consideration of the microscope time required for this high-resolution measurement makes this practically infeasible.

-Line 171: There may be a typo here as the data seem to say the opposite? Indeed there appears to be no significant difference between the lifetime of Mfd-YPet when UvrA is present but UvrB is absent (143 +/- 18 s) or when UvrA is present but UvrB is mutated in its beta-hairpin (188 +/- 46 s). The logical conclusion is therefore that UvrB mutated in its beta-hairpin cannot in fact promote disassembly of the Mfd-UvrA2-UvrB complex. Perhaps the

authors inverted the beta-hairpin and Y96A mutants in the text? This would explain the statement regarding a four-fold rate different wrt wild-type, which was measured at 18 s.

We have now clarified this point in line 214-222.

-Line 212: why is UvrA (or more precisely, UvrA-Mfd-RNAP) not indefinitely arrested in cells lacking UvrB?

Since chromosomal DNA in fast growing cells (division time: approximately 30 minutes for these strains) is acted upon by a host of DNA processing enzymes, successive rounds of replication or transcription may potentially displace Mfd-UvrA complexes. We have now pursued this question further.

In the interest of providing clarity to readers, we have now added this to the discussion (lines 280-298).

If I am not mistaken, it can be seen to reside on DNA for 97s; yet Mfd-YPet in these same conditions releases in 143 seconds. As the difference in the two numbers begins to cross the minimal threshold for statistical significance, could this be clarified?

We discovered a typographical error in the lifetime of Mfd-YPet *duvrB* – instead of 143+/-18 s, it should be 139+/- 20s. We have now corrected this in the m/s. This mistake does not influence any conclusions in this work.

Considering the uncertainties in the measurements, we have little confidence that the lifetime of UvrA-YPet in a *duvrB* strain is statistically significantly different than that of Mfd-YPet in a *duvrB* strain. Additionally, it should be noted that the measurement of UvrA-YPet lifetimes represents two UvrA-YPet species: free UvrA-YPet that can engage DNA alone (24 s lifetime), and UvrA-YPet bound to Mfd.

-Line 212: A question arose here regarding this aspect of the model. Because the Mfd-RNAP complex translocates along DNA after RNAP has been removed from damaged DNA, there is no guarantee that it is near a lesion when UvrA or UvrAB find it. Thus if UvrA-Mfd dissociate together, and because UvrA is required for loading of UvrB onto a lesion, how can downstream coupling take place if UvrA has already left the DNA?

Whereas the *in vitro* experiments beautifully demonstrate that Mfd-RNAP translocate along DNA after RNAP has been removed, currently little experimental evidence exists that this is a feature in cells carrying RNAP, Mfd, UvrA and UvrB. Supporting evidence for the relevance of this *in vitro* observation is found in our measurement that cells lacking UvrA unexpectedly exhibit Mfd-YPet lifetimes that are longer than those observed in Mfd-YPet *uvrA+* *uvrB+* cells. In this case, the lifetime of Mfd-YPet was found to be ~29 s (reported in Ho et al Nature Communications, 2018).

Assuming that the *in vitro* measurements for Mfd-RNAP translocation rate (4 nts) are valid *in vivo*, in 29s, Mfd would translocate on average 116 nt potentially allowing downstream

coupling on the length scale reported in Haines et al., 2014. If the 6 s lifetime for the UvrAB complex holds (from Fan et al), then Mfd can translocate for 12 s or 48 nt. This could place UvrAB in proximity to the lesion.

Better estimates of 1. Time taken for Mfd to reactivate/terminate transcription and 2. Wait times involved in arrival of UvrA to the site after Mfd's BHM is available for binding will enable refinement of the estimates of the length scale on which Mfd can promote downstream coupling *in vivo*.

We have now included this in the discussion to clarify the issue for the readers lines 246 – 268.

-Line 217: Please clarify why simultaneous dissociation implies strand-specific UvrB loading. R#2 also requested elaboration of our model for simultaneous dissociation. We have now expanded on this model in lines 323 – 346.

-Line 223: I returned to the paper under discussion; the release time cited was 15 +/- 3 s, not less than 10s.

Having revisited this paper we have now explicitly reproduced the details of the results as presented in the paper. Lines 269 – 279.

-Line 230-236: This section is problematic in its representation of Fan et al.: fluorescence experiments presented in Fan et al. showed that both Mfd and RNAP are released by UvrAB, essentially in a simultaneous fashion. The authors should correct their statement and model.

We have now rephrased the text (line 280-298).

-Line 237: Again, this model does not account for the continued translocation of Mfd-RNAP after remodeling of lesion-stalled RNAP (see query regarding line 212, above).

We apologize for the brevity with which the model was discussed previously leading to lack of detail. We have now significantly restructured the discussion to highlight how various aspects of the *in vitro* results may be accommodated in Mfd's functions in cells. Specifically, lines 246 – 268 now discuss the results from Mfd-YPet cells lacking UvrAB – the comparable condition where Mfd-RNAP translocate *in vivo*.

-Line 255: Similarly to line 33, there is also the 'release and catch-up' mechanism for Mfd interacting with paused RNAP, and this has not been shown to be involved in TCR. This should be mentioned.

We agree with this reviewer. We have now discussed the role of Mfd in transcription reactivation in the introduction lines 68-75.

-Line 267: Based on this model, R-loops form every time Mfd removes RNAP. Again, there should be mention of the release and catch up model, and it made clear that currently there is no evidence TCR results from Mfd interaction with paused RNAP

The authors of the release and catch-up mechanism do not comment on whether R-loops are left behind by Mfd. Further, no data are available to indicate that Mfd activity leads to R-loop formation in cells. Since we have little evidence that the release and catch-up mechanism has a relationship to R-loop formation, we hesitate to make a connection where none might exist. Certainly this is an open question for investigation.

Class II pause sites (where RNAP is backtracked) are effectively recognized by Mfd and recruit UvrAB, resulting in TCR at lesions downstream of pause sites (Haines, PNAS, 2014). We have now cited this reference to avoid confusion for readers.

-Figure 5: should be corrected: currently we only know that Mfd interacts with stalled RNAP, not paused RNAP. Sequential UvrA B loading is shown but not mentioned in the text. Order of events in figure legend should be made more clear.

The interaction of Mfd with paused RNAP has been documented in Haines et al PNAS 2014 and Le et al, Cell, 2017. Further Proshkin and Mironov (2016) demonstrate that Mfd is recruited to RNAP at sites of roadblocks.

We thank the reviewer for this comment and have now rewritten the figure legend to clarify the model for Mfd-RNAP interactions. Changes are highlighted.

Reviewers' comments:

Reviewer #1 (Remarks to the Author):

The authors have responded well to the concerns raised in the previous review, adding more experimental details, and clarifying approaches and analysis, and adding new descriptions throughout and making significant changes in the discussion. In addition, in the absence of an exogenous source of DNA damage, the authors have softened some of the terminology about stalled RNAP versus transcription-coupled repair. The latter of which can not be formally proved by their data presented, but only inferred. All together the manuscript in its present form is much stronger, reads more clearly and represents an important contribution to the field.

Reviewer #2 (Remarks to the Author):

I still find this study highly interesting and timely and would very much like to see it published in Nature Communications after minor changes. Mostly, my previous points have all been satisfactorily addressed by the authors, with exception of point 4, which still puzzles me (see below). I also still have some minor issues with clarity in the manuscript, suggestions are detailed below.

1. P.3 lines 32-45: It would help to visualize the description of Mfd domains and their roles in a schematic in Fig. 1.
2. P.9 lines 164-167: Introducing the roles of the two ATPase sites here already would help in appreciating and following the following sections. It might be worth including the recent study by Case et al. (2019 Nucleic Acids Research) here also.
3. P.9 lines 171-176 and my previous point 4: I am still confused by the effect of the presence or absence of UvrB in cells expressing the UvrA variant that cannot interact with UvrB (-/+ short lived species of Mfd on DNA) – when UvrB can also not interact with Mfd.
4. Line 187: For easier reading it would help to state here clearly (at the beginning) that it was UvrB expressing cells that gave the 52 s lifetime.
5. Line 211: I believe Figure 4a should be 4d&f.
6. Line 214: Figure 4b-c shows this?
7. Line 235: I believe Fig. 4d should be 4d&g.
8. Figure caption for Figure 2 3/f is scrambled and needs re-arranging.
9. Figure 5 caption: It would help to point out K37A and K646A where relevant.
10. Lines 269-272: The description of the experiments in this study is not very clear. Line 272 should be lifetime of RNAP not UvrA presumably. Could other components in vivo stabilize the complex, leading to faster dissociation of Mfd-RNAP in the presence of only UvrA in vitro than in vivo? In response to my previous point 9 the authors offered different concentrations in vitro and in vivo as a possible explanation for different activities. This might want to be included in the Discussion.
11. Line 299/300: It would help to state the times for UvrA and Mfd dissociation here.
12. Line 321: The authors state that the 254 nm UV light condition prioritises TCR. Does this not argue against their statement in the Introduction that no damage induction (avoiding enhanced UvrA/B expression in SOS response that would dwarf TCR) was the optimum condition for TCR detection?
13. Line 350: Why “must” a subset of their observations arise from interactions with lesion-stalled TECs, based on the fact that their Mfd-YPet construct supports TCR in cells?
14. P. 18 line 393: I am not sure if the speculation that sequestering of UvrA on DNA by Mfd and thus making it unavailable for NER of bulky lesions can be an explanation for higher mutational frequencies is not a little far-fetched. Bulky lesions are NOT TYPICALLY mutagenic.

Reviewer #3 (Remarks to the Author):

This revision is very helpful, but there remain very significant points requiring clarification, both in terms of the new changes to the manuscript and the model. The experiments described are important to numerous communities. I found the authors took some liberties overinterpreting their results and misinterpreting prior work, and this can and should be fixed. I remain confident that these issues can be addressed and the manuscript published in Nature Communications given appropriate rewording, as the issue is again not with the data but its interpretation.

The model is problematic because we cannot know what RNAP is doing when it begins to interact with the TCR machinery in these experiments. Notably, the authors' BioRxiv preprint cites a 10-second lifetime for mfd-YPet in UV-exposed cells as it works on lesion-stalled RNAP (Ghodke et al., BioRxiv 515502), while the work here cites 18 seconds, indicating that in the present manuscript Mfd is not predominantly acting on polymerase stalled at a lesion. Furthermore RNAP pauses are likely much more frequent than endogenous lesions (and not all endogenous lesions elicit a TCR response, eg 8-oxo-G). Ultimately it is not known in which ratio these are taking place. This is indeed now mentioned (line 350) but the consequences of this uncertainty are not considered in sufficient detail (see next paragraph). It is for these reasons that I suggested the authors take pains to downgrade their insistence that these are measurements on TCR-conducting complexes. It is difficult to call the process TCR when it is anybody's guess whether 0.1%, 10%, or 100% of complexes are carrying out TCR. [Contrary to the author's belief as stated in the rebuttal, Haines et al. only observe TCR-at-a-distance because the RNAP is irreversibly stalled in the first place, and not simply paused.] Although the authors state they now use the terms transcription reactivation/termination I only found a single instance of this, in the legend to Fig. 5, and an abundance of "TCR".

The distinction between paused and lesion-stalled RNAP is crucial because in vitro measurements of Le et al., carried out on paused RNAP which is then reactivated by Mfd, indicate that in this case Mfd has a roughly 30 s lifetime on DNA after reactivating RNAP. On the other hand, in vitro measurements from Graves et al. and Fan et al., carried out on stalled RNAP which cannot be reactivated by Mfd, indicate that in this case Mfd has a 1000 s+ lifetime on DNA after interacting with RNAP. Knowing whether RNAP was initially stalled or paused is clearly important as it appears to condition the stability of the resulting complex. How these distinct complexes with vastly different stabilities, and therefore likely different structures, are then acted upon by UvrA and UvrB is not known; it could be the same but it just as equally could be different. It is for these reasons that direct comparison to Fan et al. is erroneous and misleading, even though the comparison is tempting as it is the only other single-molecule assay to involve RNAP, Mfd, UvrA and UvrB.

In addition, the authors continue to fundamentally misinterpret and misrepresent Fan et al. in their new text (pg. 13, lines 269—275, pg. 13/14 lines 280—291). A first misinterpretation involves Fan et al's lifetime measurements. The published observation is that for Mfd-RNAP complexes which were seen to arrest before release by UvrA, arrest displayed stochastic behavior with a 15 s lifetime. This represented about 50% of release events. The remaining 50% of UvrA-induced release events did not display detectable arrest, but this can be quantitatively explained by the 10-s averaging time needed to distinguish translocation from arrest. Thus those complexes without detectable arrest were actually arrested but for a time shorter than 10s and continuing to obey the same single-exponential distribution – and certainly not instantly as presented here (eg line 275: "UvrA2 dissociates RNAP-Mfd from the substrate either instantly, or after a brief residence time of approximately 15 s."). This can be readily corrected.

Regarding pg. 14 (280—291), Fan's assay under discussion did not directly detect binding of UvrA to Mfd-RNAP, but detected only arrest and release of the translocating Mfd-RNAP complex by UvrA (it was not a fluorescence assay). Therefore UvrA may have been bound to the Mfd-RNAP complex for quite some time before succeeding at arresting and releasing it. Thus the result of Fan et al. regarding 15 s arrest simply cannot be interpreted here as implying it would be a 15 s dissociation time that should be detectable in the in vivo assay. In fact nothing in Fan et al. contradicts the possibility of a 140 s UvrAMfd complex. Again, this can be readily corrected.

To this reviewer, and contrary to the statements of the authors, there is a form of agreement between Fan et al and this work in that Mfd release is faster when both UvrA and UvrB are present than when UvrA alone is present. Indeed to this reviewer the authors do in fact observe dissociation of Mfd in the presence of UvrA alone, it is simply slower than when UvrB is additionally present. The data presented show that in the absence of UvrB, UvrA(K646) results in a 26 s slow lifetime for Mfd-YPet while UvrA(K37A) results in a 304 s lifetime and wt UvrA allows Mfd-YPet to be observed for 139 s. What happens to mfd-YPet at the end of these 139 s, or 26s, or 304 s – is it not dissociated? If UvrA could not dissociate Mfd-RNAP on its own, why would the lifetimes of Mfd-YPet be changing in a Δ UvrB background when changes are made to UvrA? Why does this not mean that UvrA alone can ultimately dissociate Mfd-RNAP, albeit more slowly than when UvrB is present? Although the authors may have a clear explanation in mind for this it remains rather unclear to this reviewer.

Minor remarks:

-line 34: the way this is written implies that RNAP at a pause site is terminated by Mfd, but this is incorrect as shown by Le et al. in the bump-and-catch-up model.

Response to reviewers:

We thank the reviewers for their helpful comments that continue to make this a better manuscript. We have a general comment for consideration below, and specific comments to R#2 and R#3 are presented in the point-by-point response below.

General comment:

Currently, two models are available in the literature that describe interactions between Mfd, UvrA and UvrB: First, based on elegant biochemistry, Selby and Sancar (*Science* 1993) proposed a model where "TRCF recognizes a stalled RNAP and UvrA in a UvrA₂B₁ complex and delivers UvrB to the lesion in the transcribed strand from the A₂B₁ complex and releases stalled RNAP in a sequential or highly concerted reaction". Notably, in this model, Mfd-UvrA₂ are shown to dissociate simultaneously (See steps 7 and 8 in Fig 7). This model was derived from bulk experiment conducted with Mfd (87 nM), UvrA (4nM), UvrB (120 nM), UvrC(50 nM). In this m/s, we have referred to this model as the 'classic model'.

Recently, based on single-molecule in vitro performed in the Strick lab, an alternate model for TCR has been pushed in the literature. Proponents of this model "suggest the existence of a transient UvrB-UvrA-UvrA-Mfd-RNAP complex which would convert into a UvrB-UvrA-UvrA complex upon dissociation of Mfd-RNAP (Model in Figure 4)" in Fan et al 2016. We have referred to this *decoupled handoff* as the 'contemporary' model in this paper.

In our study, we have investigated the influence of UvrAB on the residence time of Mfd on DNA after its recruitment to failed TECs *in vivo*. In Ho et al., *Nature Communications*, 2018 we demonstrated that Mfd is recruited to stalled RNAP during normal growth. In this work, we have identified unambiguously that dissociation of Mfd requires an intact proximal ATPase site of UvrA, and an intact beta-hairpin in UvrB. The model that emerges from our live cell data is that after Mfd is engaged to failed TECs, an RNAP-Mfd-UvrA₂-UvrB complex is formed in cells, and this complex disassembles in a single step where ATP hydrolysis at UvrA's proximal ATPase occurs first, followed by loading of UvrB accompanied by simultaneous dissociation of Mfd, and UvrA₂. The findings from our work completely support the classic model proposed by Selby and Sancar (1993). Our work validates this model in live cells, and further extends it by specifying catalytic and structural determinants of UvrAB that orchestrate successful and rapid handoff.

To assess whether the contemporary model (Strick lab) for Mfd-UvrAB interactions is physiologically relevant, we examined the reaction conditions under which the decoupled handoff was detected. We found that the in vitro single-molecule assays (presented in figs 1 and 2 of Fan et al 2016) were conducted using 50 – 100 pM UvrA and 250 nM UvrB (when indicated). Comparing Selby & Sancar (1993) with Fan et al., 2016, a striking difference is that the classic model appears to be operative at relatively high concentrations of UvrA (4 nM), whereas the contemporary model appears to operate at low concentration of UvrA (50-100 pM).

That UvrA functions as a dimer is settled fact (See work from Sancar, Grossman, Van Houten, Goosen laboratories). Multiple studies have reported that the K_D of dimerization for UvrA lies in the 1-10 nM range (K_D for EcUvrA[ATP] is reported to be 1.9 nM (Goosen, NAR, 37(6) 2009); BcdUvrA = 3 ± 2 nM (Van Houten lab), Orren & Sancar (1988) reported $K_d = 10$ nM). Our *in vivo* measurement of UvrA-YPet is 16 ± 4 copies per cell (this is a lower limit for the copy number of UvrA under our growth conditions, arising from poor translation of *uvrA-ypet*) translating to ~ 6 nM (typical cells have the dimensions: length = ~ 5.5 μm , width = 0.5 μm). In Ghodke et al bioRxiv 2019, we demonstrate that a plasmid expressing 120 copies of UvrA-YPet per cell (46 nM) exhibits UV survival indistinguishable to wild-type, suggesting that the concentration of wildtype untagged UvrA is similar. Other estimates of UvrA concentrations are consistent with our measurements (Stracy et al, 2016).

One can estimate the percentage of UvrA that is dimerized using the published values for the equilibrium dissociation constant K_D for EcUvrA in the presence of ATP (1.9 nM, Goosen NAR 2009). At a total UvrA-YPet concentration of 6 nM (*in vivo*), the fraction of UvrA found as dimers is 66%, while at a concentration wildtype UvrA of 46 nM, 86% of the UvrA is dimerized. In contrast, at 100 pM only ~ 8.8 % of the UvrA exists as a dimer. This value is lower (4.8%) at UvrA concentration of 50 pM.

Therefore, we believe that the differences between the 'decoupled' handoff seen by Strick and co-workers in the presence of UvrA alone, and observations by Selby and Sancar as well as our own *in vivo* observations can be adequately explained based on the differences in UvrA concentration. Simply put, whereas the *in vivo* conditions reflect conditions in which UvrA operates as a dimer, the single molecule *in vitro* work used to support the contemporary model was performed under conditions where at best 8% of the population of UvrA is dimerized. Unfortunately, no evidence by way of direct visualization of UvrA is provided by Fan et al., 2016 to reassure readers that the data reflect conditions where dimeric UvrA is the species responsible for dissociating Mfd in the presented trajectories. On the other hand, formation of dimeric UvrA is a prerequisite for detection of UvrB facilitated dissociation of Mfd observed in our study. The beauty of the two different experimental conditions (our work+Selby and Sancar 1993 vs. Fan et al 2016) is that they reveal the various ways in which molecular handoff may occur, depending on the stoichiometry of UvrA. Upon request by R#2, have now added this explanation in the 'Discussion' in this paper.

For these reasons, in disagreement with R#3's position that no comparisons should be made between our work and Fan et al, 2016, we believe that the leading *in vitro* model for Mfd-UvrAB interactions in the context of TCR deserves to be scrutinized and discussed. In doing so, one can uncover new properties of molecular handoff in DNA repair. The decoupled handoff presented in the contemporary model does not appear to primarily operate in live cells under our growth conditions, *even if it is readily detected in vitro*. The findings in our work unambiguously reveal that under the physiologically relevant conditions, a UvrB mediated facilitated dissociation of Mfd operates predominantly.

Additionally, we have made minor editorial changes to improve the readability of the text (all highlighted in yellow).

Reviewer #1 (Remarks to the Author):

The authors have responded well to the concerns raised in the previous review, adding more experimental details, and clarifying approaches and analysis, and adding new descriptions throughout and making significant changes in the discussion. In addition, in the absence of an exogenous source of DNA damage, the authors have softened some of the terminology about stalled RNAP versus transcription-coupled repair. The latter of which can not be formally proved by their data presented, but only inferred. All together the manuscript in its present form is much stronger, reads more clearly and represents an important contribution to the field.

Reviewer #2 (Remarks to the Author):

I still find this study highly interesting and timely and would very much like to see it published in Nature Communications after minor changes. Mostly, my previous points have all been satisfactorily addressed by the authors, with exception of point 4, which still puzzles me (see below). I also still have some minor issues with clarity in the manuscript, suggestions are detailed below.

1. P.3 lines 32-45: It would help to visualize the description of Mfd domains and their roles in a schematic in Fig. 1.

We have now added a new panel to figure 1 to describe the apo-structure of Mfd (Deaconescu 2006) and indicated the various domains in Fig 1a.

2. P.9 lines 164-167: Introducing the roles of the two ATPase sites here already would help in appreciating and following the following sections. It might be worth including the recent study by Case et al. (2019 Nucleic Acids Research) here also.

Reference to Case et al. 2019 is now added in line 159.

3. P.9 lines 171-176 and my previous point 4: I am still confused by the effect of the presence or absence of UvrB in cells expressing the UvrA variant that cannot interact with UvrB (-/+ short lived species of Mfd on DNA) – when UvrB can also not interact with Mfd.

The UvrA(K646A) mutant is defective in interacting with UvrB, but still possesses a residual ability to interact with UvrB (as demonstrated in Myles et al. 1991, Manelyte et al. 2010). Since UvrB does not interact with Mfd, the increased lifetime of Mfd in the presence/absence of UvrB must reflect additional stabilization by UvrB in residual interactions with mutant UvrAs. No structural evidence is available to confirm this hypothesis.

4. Line 187: For easier reading it would help to state here clearly (at the beginning) that it was UvrB expressing cells that gave the 52 s lifetime.

This is now added in line 185.

5. Line 211: I believe Figure 4a should be 4d&f.

This is now fixed (line 209).

6. Line 214: Figure 4b-c shows this?

This is now fixed (line 212).

7. Line 235: I believe Fig. 4d should be 4d&g.

This is now fixed (line 233).

8. Figure caption for Figure 2 3/f is scrambled and needs re-arranging.

This is now fixed.

9. Figure 5 caption: It would help to point out K37A and K646A where relevant.

This is now fixed (lines 822, 825).

10. Lines 269-272: The description of the experiments in this study is not very clear. Line 272 should be lifetime of RNAP not UvrA presumably. Could other components in vivo stabilize the complex, leading to faster dissociation of Mfd-RNAP in the presence of only UvrA in vitro than in vivo? In response to my previous point 9 the authors offered different concentrations in vitro and in vivo as a possible explanation for different activities. This might want to be included in the Discussion.

At the request of this reviewer, we have now added elaborated on this potentially explanation in this discussion (lines 300 -325).

11. Line 299/300: It would help to state the times for UvrA and Mfd dissociation here.

This is now added (line 268-269).

12. Line 321: The authors state that the 254 nm UV light condition prioritises TCR. Does this not argue against their statement in the Introduction that no damage induction (avoiding enhanced UvrA/B expression in SOS response that would dwarf TCR) was the optimum condition for TCR detection?

Detection of the TCR signal is challenging in *recA*⁺ cells since SOS induction leads to upregulation of UvrAB protein expression and consequently drowns out any influence of Mfd on survival after UV in traditional UV survival assays (see Selby and Sancar, Science,

1993). For these reasons, it has been argued by others that detectability of TCR should be better in normal growing cells, since the GGR pathway will be operational to a much lesser extent. In the introduction, we have merely summed up current opinion on the best conditions for detecting TCR activity.

The statement that TCR is prioritized in cells exposed to 254 nm UV light reflects data by Crowley and Hanawalt, where using rif, the authors demonstrate that TCR of CPDs is indeed prioritized in the UV treated condition. We have now qualified this statement as '... a condition in which the repair of cyclobutane pyrimidine dimers via TCR is prioritized') in line 297-298.

13. Line 350: Why "must" a subset of their observations arise from interactions with lesion-stalled TECs, based on the fact that their Mfd-YPet construct supports TCR in cells?

This work is predicated on three axioms:

1. Cellular metabolism results in endogenous DNA damage during normal growth. (Due to technical limitations, we cannot formally quantify and describe the spectrum of DNA damage in cells growing in our flow cell.)
2. RNAP is stalled at metabolic lesions during normal growth
3. Mfd is recruited to these stalled RNAP (among other types of failed TECs) and TCR follows

We note that the Mfd-YPet construct supports TCR in cells in response to UV-irradiation (Ho et al., *Nature Communications*, 2018)

Therefore, we hypothesized that *in a non-zero set of events, Mfd-YPet is recruited to the site of RNAP stalled at endogenous lesions, and can support TCR at these sites.*

For this reason, we stated that '... a subset of the observations in our experiments must arise from interactions with lesion-stalled TECs'. Because we cannot quantify the fraction of events at which TCR occurs in our growth condition, we have now rephrased this to 'may' (line 353).

14. P. 18 line 393: I am not sure if the speculation that sequestering of UvrA on DNA by Mfd and thus making it unavailable for NER of bulky lesions can be an explanation for higher mutational frequencies is not a little far-fetched. Bulky lesions are NOT TYPICALLY mutagenic.

Bulky lesions are not mutagenic *because* they are corrected by NER in an error-free manner. In the event that UvrAB are sequestered at sites of Mfd-RNAP, replication forks encountering such lesions may utilize error-prone polymerases to tolerate such lesions via

TLS, potentially in the context of SOS. We agree with this reviewer that this speculation needs to be tested experimentally.

Reviewer #3 (Remarks to the Author):

This revision is very helpful, but there remain very significant points requiring clarification, both in terms of the new changes to the manuscript and the model. The experiments described are important to numerous communities. I found the authors took some liberties overinterpreting their results and misinterpreting prior work, and this can and should be fixed. I remain confident that these issues can be addressed and the manuscript published in Nature Communications given appropriate rewording, as the issue is again not with the data but its interpretation.

The model is problematic because we cannot know what RNAP is doing when it begins to interact with the TCR machinery in these experiments.

It appears that this reviewer has misunderstood the scope of the model presented here. In agreement with this reviewer, we note that our work does not allow us to comment on two factors (enlisted below) that would have enabled us to *conclude* that the observations correspond to TCR

1. The precise cause for RNAP stalling/pausing and
2. Whether downstream repair machinery (UvrCD, Lig and Pol) are subsequently recruited to the sites visualized here, and consequently whether the processes described here are exclusively reflecting DNA repair.

For these reasons, we have restricted the scope of the model to describe the observed interactions between the transcription and repair machineries ie, Mfd, UvrA and UvrB *after* Mfd is recruited to a failed TEC.

Currently, it is known that UvrAB are recruited to Mfd during TCR. On the other hand, it is not known whether UvrAB are recruited to Mfd engaged in transcription reactivation. Given currently published data, we can neither include nor exclude the possibility that UvrAB are involved in transcription reactivation. However, both reactions are accommodated in the model presented here. Detailed investigations regarding transcription reactivation vs termination remain to be performed, primarily due to technical limitations.

We have now made changes to figure 5 describing the model to avoid conveying to readers that the cause of TEC failure is known.

Notably, the authors' BioRxiv preprint cites a 10-second lifetime for mfd-YPet in UV-exposed cells as it works on lesion-stalled RNAP (Ghodke et al., BioRxiv 515502), while the work here cites 18 seconds, indicating that in the present manuscript Mfd is not predominantly acting on polymerase stalled at a lesion.

The claim that 'Mfd is not predominantly acting on polymerase stalled at a lesion' appears to be based on a naïve interpretation of the mean lifetimes of Mfd detected in UV treated

cells. Here the referee seems to assume that the 12 s lifetime corresponds only to TCR at lesions. We suggest that the data should be considered more carefully prior to reaching the conclusion that the lifetime of Mfd at sites of UV lesions is 12 s (not 10 s as the referee notes).

Note that in UV treated cells the lifetime of Mfd drops from ~14s at 25 min after UV to 8 s at 100 min after UV (presenting as the average ~12 s lifetime cited by the referee). Notably, consistent with participation in TCR, UvrA's lifetimes match those of Mfd in UV-treated cells (these values are available in Ghodke bioRxiv 2019).

It is true that in the absence of UV damage the lifetime of Mfd-Ypet is ~18 s (Ho et al., 2018). It should be noted that at modestly enhanced expression levels of UvrB, the lifetime of Mfd-YPet drops from ~18s to ~11s (Figure 4 of this m/s) *even in the absence of exogenous DNA damage*. Clearly, UvrB concentration is a rate limiting factor for turnover of Mfd in cells.

In light of these data, it is therefore non trivial to conclude that the lifetime of Mfd at lesions is ~11 s, and ~18s lifetimes represents participation in non-TCR related processes. If the referee's argument Mfd engaged in TCR possesses a lifetime of ~12 s, then, by extension it follows that the ~11 s lifetime (detected at higher expression levels of UvrB in normal growing cells) must also reflect TCR (contradicting their own comments that TCR is not detected in our cells in normal growth – see below, and in previous rounds of review!).

We disagree with this naive interpretation and present the following argument for consideration. The simplest model that accommodates all three observations is that the observed lifetime of Mfd-YPet reflects (at least) two processes – one where the RNAP-bound Mfd complex awaits the arrival of UvrA or UvrB (this lifetime would be sensitive to UvrB copy numbers in cells) and a second process that involves the UvrA ATPase mediated loading of UvrB on DNA that triggers dissociation of Mfd (described here).

Consistent with this model, at normal cellular expression levels of UvrB, Mfd-RNAP resides on DNA longer (~18s) since UvrB copy number is lower and these are required to turn over not just Mfd-UvrA₂, but also UvrA₂ complexes on DNA. At higher expression levels, the enhanced copy numbers of UvrB efficiently turn over Mfd (~11s in the absence of exogenous damage) by the facilitated dissociation mechanism identified here. Under conditions of SOS, UvrB levels are thought to be five-fold higher, consistent with the quicker turnover of both UvrA and Mfd (findings presented in Ghodke et al, 2019).

We have now summarized this comment in lines 235 -241 to provide clarity on this issue.

Furthermore, RNAP pauses are likely much more frequent than endogenous lesions (and not all endogenous lesions elicit a TCR response, eg 8-oxo-G). Ultimately it is not known in which ratio these are taking place. This is indeed now mentioned (line 350) but the consequences of this uncertainty are not considered in sufficient detail (see next

paragraph). It is for these reasons that I suggested the authors take pains to downgrade their insistence that these are measurements on TCR-conducting complexes. It is difficult to call the process TCR when it is anybody's guess whether 0.1%, 10%, or 100% of complexes are carrying out TCR. [Contrary to the author's belief as stated in the rebuttal, Haines et al. only observe TCR-at-a-distance because the RNAP is irreversibly stalled in the first place, and not simply paused.] Although the authors state they now use the terms transcription reactivation/termination I only found a single instance of this, in the legend to Fig. 5, and an abundance of "TCR".

We agree entirely with this referees argument, and for these reasons we have titled the paper: "Single-molecule imaging reveals molecular coupling between transcription and DNA repair machinery in live cells" with no reference to "Transcription-coupled repair".

We have used the term TCR very carefully in this m/s. There are 18 instances of the use of the term TCR in this m/s, 16 of which are used to provide introduction to either NER or Mfd and exactly two instances where the implications of our work impinge on TCR. In all these uses, we have taken great care to acknowledge that the handoff complexes described here can form at TECs that can be merely paused, or irreversibly stalled at lesions, or roadblocks, and consequently. We understand that we cannot ascribe the observations exclusively to TCR or reactivation and have reiterated these points several times in the m/s so that readers can judge the validity and relationship of our data to the TCR reaction.

The egregious demands to censor the use of the term TCR do a great disservice to readers, and for this reason we have elected to retain the use of the term TCR where appropriate (as described above).

The distinction between paused and lesion-stalled RNAP is crucial because in vitro measurements of Le et al., carried out on paused RNAP which is then reactivated by Mfd, indicate that in this case Mfd has a roughly 30 s lifetime on DNA after reactivating RNAP. On the other hand, in vitro measurements from Graves et al. and Fan et al., carried out on stalled RNAP which cannot be reactivated by Mfd, indicate that in this case Mfd has a 1000 s+ lifetime on DNA after interacting with RNAP. Knowing whether RNAP was initially stalled or paused is clearly important as it appears to condition the stability of the resulting complex. How these distinct complexes with vastly different stabilities, and therefore likely different structures, are then acted upon by UvrA and UvrB is not known; it could be the same but it just as equally could be different. It is for these reasons that direct comparison to Fan et al. is erroneous and misleading, even though the comparison is tempting as it is the only other single-molecule assay to involve RNAP, Mfd, UvrA and UvrB.

We agree with this referee that whether paused and stalled RNAP have different kinetic properties when acted upon by Mfd is an important question. As noted several times, the data presented in this work do not allow us to comment on whether the RNAP's engaged by Mfd are simply paused, or stalled at lesions/roadblocks.

All models discussed here concern themselves with activities that occur once Mfd arrives at the scene of a failed TEC, be it paused, or irreversibly stalled at lesions/roadblocks. Since we cannot detect the cause of transcription failure, we have discussed implications of our findings on each of the various types of failed TECs in lines 350 – 390, ie, at sites of lesions (outcome being TCR), sites of class II pause sites and roadblocks and pause sites at which transcription can be reactivated. Finally, we have noted that these questions await further investigation and are currently out of the scope of this m/s due to technical limitations.

We do not believe direct comparison to Fan et al is erroneous or misleading. We reiterate that we do not believe that only TCR is captured in these data. TCR is only one potential outcome from the interactions detected in our study, and hence deserves to be discussed. Avoiding comparisons to other investigations of handoff complexes formed by RNAP-Mfd-UvrA and UvrB would be a great disservice to readers in the fields of DNA repair and transcription who might be interested in understanding how these proteins function *in vivo*.

In addition, the authors continue to fundamentally misinterpret and misrepresent Fan et al. in their new text (pg. 13, lines 269—275, pg. 13/14 lines 280—291). A first misinterpretation involves Fan et al's lifetime measurements. The published observation is that for Mfd-RNAP complexes which were seen to arrest before release by UvrA, arrest displayed stochastic behavior with a 15 s lifetime. This represented about 50% of release events. The remaining 50% of UvrA-induced release events did not display detectable arrest, but this can be quantitatively explained by the 10-s averaging time needed to distinguish translocation from arrest. Thus those complexes without detectable arrest were actually arrested but for a time shorter than 10s and continuing to obey the same single-exponential distribution – and certainly not instantly as presented here (eg line 275: "UvrA2 dissociates RNAP-Mfd from the substrate either instantly, or after a brief residence time of approximately 15 s."). This can be readily corrected.

We apologize for the unintentional misrepresentation. We have now replaced this discussion with "...the lifetime of the arrested Mfd-UvrA complex has been previously measured to be 15s *in vitro*" (line 273).

Regarding pg. 14 (280—291), Fan's assay under discussion did not directly detect binding of UvrA to Mfd-RNAP, but detected only arrest and release of the translocating Mfd-RNAP complex by UvrA (it was not a fluorescence assay). Therefore UvrA may have been bound to the Mfd-RNAP complex for quite some time before succeeding at arresting and releasing it. Thus the result of Fan et al. regarding 15 s arrest simply cannot be interpreted here as implying it would be a 15 s dissociation time that should be detectable in the *in vivo* assay. In fact nothing in Fan et al. contradicts the possibility of a 140 s UvrAMfd complex. Again, this can be readily corrected.

We completely agree with this reviewer here – the two assays report on different activities. This is now added in line 274.

To this reviewer, and contrary to the statements of the authors, there is a form of agreement between Fan et al and this work in that Mfd release is faster when both UvrA and UvrB are present than when UvrA alone is present. Indeed to this reviewer the authors do in fact observe dissociation of Mfd in the presence of UvrA alone, it is simply slower than when UvrB is additionally present.

This statement is only partially correct because it fails to incorporate findings from our work (below) and to that extent, it is incomplete.

In our assays, lifetime of Mfd-YPet in $\Delta uvrB$ cells is 139 ± 20 s (this work), whereas that of UvrA-YPet in $\Delta uvrB$ cells is 97 ± 18 s (Ghodke bioRxiv 2019). Additionally, in Ghodke 2019, we provide evidence that Mfd and UvrA associate in cells both in the absence, and presence of DNA damage. Further, in both cases, beta-hairpin mutants of UvrB arrest Mfd and UvrA with comparable lifetimes. The only explanation that accommodates all these observations is that Mfd-UvrA form a complex, and in the presence of wild-type UvrB, both Mfd and UvrA dissociate together.

Therefore, the statement that 'the authors do in fact observe dissociation of Mfd in the presence of UvrA alone, it is simply slower than when UvrB is additionally present' while correct does not capture the reality in its entirety and is therefore incomplete.

It is more appropriate to state that both Mfd and UvrA₂ dissociate simultaneously in the absence of UvrB (here the lifetimes are between 97-140 s), whereas in the presence of UvrB, disassembly of Mfd-UvrA₂ is faster on the order of 11s - 18s, depending on the expression level of UvrB.

This statement has been extensively elaborated in the discussion section (lines 268 – 299). Changes are highlighted in yellow.

The data presented show that in the absence of UvrB, UvrA(K646) results in a 26 s slow lifetime for Mfd-YPet while UvrA(K37A) results in a 304 s lifetime and wt UvrA allows Mfd-YPet to be observed for 139 s. What happens to mfd-YPet at the end of these 139 s, or 26s, or 304 s – is it not dissociated?

These numbers represent mean lifetimes measured from a dataset with $>10^5$ observations. So yes, for an "average" molecule, Mfd-YPet would dissociate at the end of these residence times (this has been indicated in figure 5).

If UvrA could not dissociate Mfd-RNAP on its own, why would the lifetimes of Mfd-YPet be changing in a $\Delta UvrB$ background when changes are made to UvrA?

This referee is asking why the lifetimes of DNA- and RNAP-bound Mfd-Ypet in *mfd-yPET uvrA⁺ $\Delta uvrB$* cells (~139 s; single species), *mfd-yPET uvrA(K646A) $\Delta uvrB$* cells (slow lifetime ~26 s; fast lifetime = 1.1 s) and *mfd-yPET uvrA(K37A) $\Delta uvrB$* cells (~304 s; single species) are different.

The distal ATPase (K646A) mutant of UvrA engages weakly with UvrB possessing only 1% UvrB loading activity compared to wildtype (Myles...Sancar, 1991). Since Mfd and UvrB interact with UvrA via the same interface, we expect that UvrA(K646A) will also engage weakly with Mfd. In this case, we expect Mfd-YPet to behave like it does in the absence of UvrA. Indeed, Mfd-YPet has a lifetime of ~29 s in cells. Whereas, Mfd-YPet in the UvrA(K646A) background exhibits a lifetime of ~26 s. We interpreted this species as the translocating DNA bound complex of Mfd that occurs after initial engagement with RNAP.

Previously, Myles et al., 1991 showed that UvrA(K37A) possesses 10% of the UvrB loading activity compared to wildtype UvrA. Further, Wagner et al., 2010 showed that this mutant can form the UvrA(K37A)UvrB complex on DNA but is impaired in efficient loading of UvrB, due to impaired ATP hydrolysis. Our studies with beta-hairpin mutants of UvrB demonstrate that loading of UvrB is critical for dissociation of Mfd and UvrA. Therefore, the *mfd-YPet uvrA(K37A)* strain resembles the *mfd-YPet ΔuvrB* strain, in that loading of UvrB does not occur in both strains. Consistent with this, Mfd-YPet is arrested for 139s and 304 s respectively. The faster dissociation of Mfd-YPet in the *ΔuvrB* expressing wild-type UvrA is consistent with a scenario where ATP hydrolysis by UvrA at the proximal site (K37) is able to dissociate the (RNAP)-Mfd-UvrA₂ complex albeit with a rate constant that is approximately ten-fold slower than that observed in the presence of UvrB.

The roles of ATP hydrolysis by UvrA in loading UvrB or Mfd are currently poorly understood. To our knowledge, no other experiments performed with complementary techniques are available in the literature that enable us to conclusively identify how the (RNAP)-Mfd-UvrA₂-DNA complex differs from the (RNAP)-Mfd-UvrA(K646A)₂-DNA, or (RNAP)-Mfd-UvrA(K37A)₂-DNA complexes.

Based on these data we have proposed a model in which ATP hydrolysis at the proximal ATPase, and the ability of UvrB to engage DNA are together necessary and sufficient to complete the handover within 18s. As the data show, if these two conditions are not met, the rate of reaction is slower.

Why does this not mean that UvrA alone can ultimately dissociate Mfd-RNAP, albeit more slowly than when UvrB is present? Although the authors may have a clear explanation in mind for this it remains rather unclear to this reviewer.

The central goal of this work is to identify which of the following reactions occur in cells:

Classic model at sites of RNAP (Sancar):

Ia: Mfd-UvrA-UvrA-DNA → Mfd + UvrA-UvrA + DNA

Ib: Mfd-UvrA-UvrA-UvrB-DNA → Mfd + UvrA-UvrA + DNA-UvrB

Contemporary model (Strick):

IIa: Mfd-UvrA-UvrA-DNA → Mfd + DNA-UvrA-UvrA

IIb: Mfd-UvrA-UvrA-UvrB-DNA → Mfd + DNA-UvrA-UvrA-UvrB

The key difference between the two models being whether UvrA dissociates along with Mfd (classic model) or stays behind (contemporary model).

We apologize if we do not understand this referee's question but we interpret the reviewer's request to mean: how we can rule out that IIa does not occur in cells.

1. The *in vivo* approach presented here (and published in Ho et al., Biophysical Journal 2019) has the ability to detect binding lifetimes in the range from 100 ms to ~300 s.

2. When UvrB is absent, we measured the lifetime of Mfd-YPet to be 139 s. In Ghodke et al, bioRxiv 2019 (see table 1 in that paper) we showed that the lifetime of UvrA-YPet in *uvrA-ypet mfd+ ΔuvrB* cells is 97 ± 18 s (single species), and in *uvrA-ypet Δmfd ΔuvrB* cells is 22 ± 8 s (chr) or 24 ± 1 s (low copy plasmid). Thus in the absence of UvrB, *both Mfd and UvrA are arrested on DNA*, suggesting that they might work as part of the same complex.

3. Further, using a β -hairpin deletion mutant of UvrB we found that UvrA-YPet is arrested on DNA with a lifetime of 148 ± 36 s. In this same background, Mfd-YPet is arrested on DNA with a lifetime of 188 ± 46 s.

4. Upon irradiation, the lifetimes of Mfd and UvrA *in vivo* are essentially identical (see Figure 5 d in Ghodke bioRxiv 2019 515502), further lending support to the hypothesis that Mfd-UvrA form a complex in the absence of UvrB.

5. Neither in the *uvrA-YPet ΔuvrB* dataset, nor in the *mfd-YPet ΔuvrB* dataset, do we detect a species that dissociates on a timescale comparable to that in *uvrB+* cells (Mfd-YPet: 18s; UvrA-YPet: 12-18s).

If UvrA alone can dissociate Mfd as the reviewer proposes (IIa), we would not have detected an influence of UvrB on the lifetime of Mfd. As is unambiguously clear from our data, the absence of UvrB or in the presence of loading mutants of UvrB, Mfd is arrested on DNA on a timescale that is 10-15 fold longer than in the presence of wildtype UvrB.

Thus, we have argued that under conditions where our experimental approach permits robust detection of lifetimes in the 0.1 – ~300 s, there is no indication that Mfd dissociates on a different timescale in cells lacking UvrB. In UV-treated cells, the lifetimes of Mfd and UvrA are identical. Further, the β -hairpin of UvrA governs dissociation of both Mfd, and UvrA. These data overwhelmingly support reactions outlined in Ia and Ib.

The model emerges that Mfd-UvrA form part of a complex in which UvrB individually regulates the lifetimes of both factors. We have therefore proposed the model in this paper that the DNA-bound Mfd-UvrA₂-UvrB complex disassembles in a single step leaving behind UvrB at the site. This model is identical to the model proposed by Selby and Sancar.

Minor remarks:

-line 34: the way this is written implies that RNAP at a pause site is terminated by Mfd, but this is incorrect as shown by Le et al. in the bump-and-catch-up model.

In the interest of clarity we have now removed the mention of pauses in line 34 and restricted discussion to 68-75 in the introduction.

Reviewers' comments:

Reviewer #1 (Remarks to the Author):

The authors have responded well to the concerns raised in the revision. They make a fairly strong argument regarding differences in concentrations in their and the Strike paper, thus and because of these concentration differences the two results do not have to be completely congruent. It is important make this new study available to the research community.

Reviewers' comments:

Reviewer #3 (Remarks to the Author):

The manuscript is meritorious and timely in terms of its novel data, however to this reviewer the authors' model-building leaves much to be desired. Their effort is not made easier by the absence of quantitative information regarding the exact nature of the complexes being studied (ie fraction of complexes formed on paused RNAP, damage-stalled RNAP, etc.). In view of this it seems natural to suggest the authors temper the tone taken in discussing the work from Strick and coworkers; it is not necessary for the authors to get their point across and the comparisons are often unconvincing because the experiments measure different things.

— The authors repeatedly turn to the historical model of Selby and Sancar (eg line 115, 318, Figure 1...), in which Mfd-UvrA2 are drawn dissociating simultaneously. Unfortunately this reviewer fails to see any experimental evidence whatsoever for this assertion in the text of the 1993 manuscript. It seems this aspect of the model is entirely speculative and lacks supporting data, even in the subsequent years (it is termed 'plausible' in Selby and Sancar's own words). If the authors have a page or figure number or reference they believe this reviewer has missed they should let this reviewer know. Reference [48] regarding UvrB-mediated dissociation of UvrA is irrelevant here as it does not include Mfd in its scope. Therefore this reviewer feels the Selby and Sancar model should not be used in this manner 26 years later, considering that simultaneous Mfd-UvrA2 dissociation has still yet to be demonstrated and that other major elements of the model have been updated. As we see in step 5 of the Sancar model for example – they didn't know if RNAP left the system at this point, but it was the best guess at the time with the available data; we now know RNAP stays attached to Mfd as part of a DNA-Mfd-RNAP complex which translocates along DNA. This is for instance lost on readers if the authors now decide to claim the "classic model" is correct as they draw in their model Figure 1. Furthermore there is every indication from not just one but a multitude of single-molecule assays (Howan et al. Nature 2012 and Graves et al., NSMB 2016, in addition to Fan et al.) that removal of Mfd-RNAP from DNA is accomplished by UvrA/UvrB. So the authors must negate extensive work to come down so strongly for the "classic" model as their Figure 1 and overall message convey. This comparison of models is an ineffective narrative through which to explain these findings.

— Regarding the nearly full page of new text aimed at the findings of Fan et al., this reviewer absolutely disagrees with the misleadingly formulated statement that "Strick and colleagues failed to detect coordinated handoff" (line 322). The authors cannot ignore that no experiments in Fan et al. were designed to directly test coordinated handoff as no fluorescent UvrA or UvrB were used, so how could they have succeeded or failed at detecting coordinated handoff? If the authors take for proof of this failure the fact that UvrA alone is able to remove Mfd-RNAP, this reviewer notes that the authors are themselves finally acknowledging that UvrA on its own may be able to remove Mfd-RNAP, albeit slowly (see below). If the authors take for proof of this failure that no proteins were observed on the DNA after UvrA/UvrB removal of Mfd-RNAP, that is only because there were no fluorescent UvrA or fluorescent UvrB proteins were used in Fan et al. This straw man representation of other people's work is unacceptable, and it must be pointed out that this is the umpteenth time in this review process that major errors in the authors' presentation of Fan et al. have had to be corrected. Although

reexamination of prior work is key to the scientific process, such reexamination should not be done in the partial way that the authors have allowed themselves in their reading of Fan et al.

In general it is surprising to see the treatment of Fan et al. in this work. Fan et al only showed that UvrAB could provoke release of Mfd-RNAP, and that UvrA could also, albeit more slowly and with reduced efficiency. We note that, as is now acknowledged in this round of review, the authors are also observing this activity of UvrA alone (K37A vs WT). The fact that Fan et al. observed UvrA releasing Mfd-RNAP without UvrB does not mean it is frequent in vivo where UvrB is present. Again, for Fan et al.'s experiment carried out in the presence of UvrB, the experiment does not directly test coordinated handoff, and the inference in Fan's model that UvrA2B1 was retained on DNA was again a speculation made by the authors (as was done in the 1993 model by Selby and Sancar). Again model comparison is an unattractive means through which this paper could be presented. Is it not preferable to focus on your interesting findings and their biological significance, rather than focus on which model it backs up the most?

— The new argument that Fan et al.'s use of 50-100 pM concentration of UvrA leads the authors to "expect that disassembly of the handoff complex via UvrB-facilitated dissociation of Mfd and UvrA would be exceedingly rare" is flawed and certain to mislead readers. The authors state correctly in their point-by-point reply that when present in Fan et al.'s assays UvrB concentration was 250 nM, but this is missing in the manuscript's discussion (line 320). The authors also state correctly that it is widely accepted that UvrA acts as a dimer. Given UvrA's dissociation constant this is practically a necessary assumption for all assays given that there is always a mix of monomer and dimer in everybody's experiments. Low UvrA2 dimer concentration will only affect the time taken for UvrA₂/UvrB to bind Mfd-RNAP. This can even be a very slow process given the stability of Mfd-RNAP on DNA. If UvrA does not already come to Mfd-RNAP with UvrB, the time it takes UvrB to join the complex and carry out handoff is only limited by the concentration of Mfd-RNAP, and is independent of the concentration of UvrA. It is therefore unclear why low dimeric UvrA would lead to exceedingly rare handoff via UvrB, apart from the diffusion-based rate-limiting effect of low dimer concentration.

— In their rebuttal the authors acknowledge now regarding UvrA(K37A) that **"The faster dissociation of Mfd-YPet in the ΔUvrB expressing wild-type UvrA is consistent with a scenario where ATP hydrolysis by UvrA at the proximal site (K37) is able to dissociate the (RNAP)-Mfd-UvrA2 complex albeit with a rate constant that is approximately ten-fold slower than that observed in the presence of UvrB."** This statement, which offers a path for agreement between these data and those of Fan et al., needs to be incorporated into the manuscript's discussion at an appropriate place. This could be accomplished for instance by correcting the text around lines 288-290 to read "...suggests that this ATPase activity alone is sufficient for dissociation of Mfd, albeit at an approximately tenfold slower rate than that observed in the presence of UvrB." It would even be perhaps useful point out that this observation helps reconcile the authors' results with those of Fan et al.

— The correction made line 275 is too vague to be of use to readers and needs to be improved. The authors now write "Whereas the lifetime of the arrested Mfd-UvrA complex has been

previously measured to be 15s in vitro, a short-lived complex of Mfd-YPet with a lifetime of 15s was not detected in UvrB- cells. Potentially the two experiments measure different activities.” This reviewer realizes the first sentence introduces confusion between Fan et al.’s use of the term arrested (in which it means that the translocating Mfd-RNAP complex has ceased translocating) and the use the authors make of it for their own experiments (protein localized to DNA rather than diffusing). Of course the assays measure entirely different activities, the authors acknowledged them in their rebuttal but do not clarify here. If the authors insist on making such comparisons, they should correct it here. The bottom line is that the assay of Fan et al. being discussed here does not make any claim as to the amount of time the UvrA-Mfd-RNAP complex exists on DNA before Mfd-RNAP is arrested in its translocation (present or absent UvrB), and so the time they do measure cannot be held up for comparison to what is observed in this manuscript.

— Line 366, this reviewer does not understand the authors’ assertion that symmetry-breaking is a problem in the model of Fan et al. On one hand if a UvrA2-UvrB1 complex is deposited on DNA after interaction with Mfd-RNAP then this is precisely a symmetry-breaking complex. In terms of composition it will be stable over longer periods of time the harder it is for a DNA-bound complex to acquire new subunits (currently not known). On the other hand If the authors’ point is that the Fan model claims UvrA alone to be the TCR taskmaster then there is a misrepresentation of Fan et al. Although Fan et al.’s experiments showed UvrA alone is capable of removing Mfd-RNAP, there are no claims made about how frequent this is in vivo, in particular given the presence of UvrB, and the bottom line of Fan et al has always been that UvrAB is the key to rapidly moving the TCR process towards success.

General comments:

We appreciate R#3's efforts to enhance this work by helping us avoid misrepresentation of Strick's findings. We have now removed any confusing statements that raised concerns for R#3 and hope that we have satisfactorily addressed these concerns.

We have additionally changed Figure 1 to indicate a simple model for recruitment of UvrAB to Mfd that summarizes previous work.

Reviewer #1 (Remarks to the Author):

The authors have responded well to the concerns raised in the revision. They make a fairly strong argument regarding differences in concentrations in their and the Strike paper, thus and because of these concentration differences the two results do not have to be completely congruent. It is important make this new study available to the research community.

We thank the reviewer for their positive evaluation of this work. To avoid confusing readers and distracting readers from the main findings of our work and in light of concerns raised by R#3, we have removed this section from the discussion.

Reviewer#3

The manuscript is meritorious and timely in terms of its novel data, however to this reviewer the authors' model-building leaves much to be desired. Their effort is not made easier by the absence of quantitative information regarding the exact nature of the complexes being studied (ie fraction of complexes formed on paused RNAP, damage-stalled RNAP, etc.). In view of this it seems natural to suggest the authors temper the tone taken in discussing the work from Strick and coworkers; it is not necessary for the authors to get their point across and the comparisons are often unconvincing because the experiments measure different things. —

1. The authors repeatedly turn to the historical model of Selby and Sancar (eg line 115, 318, Figure 1...), in which Mfd-UvrA2 are drawn dissociating simultaneously. Unfortunately this reviewer fails to see any experimental evidence whatsoever for this assertion in the text of the 1993 manuscript. It seems this aspect of the model is entirely speculative and lacks supporting data, even in the subsequent years (it is termed 'plausible' in Selby and Sancar's own words). If the authors have a page or figure number or reference they believe this reviewer has missed they should let this reviewer know. Reference [48] regarding UvrB-mediated dissociation of UvrA is irrelevant here as it does not include Mfd in its scope. Therefore this reviewer feels the Selby and Sancar model should not be used in this manner 26 years later, considering that simultaneous Mfd-UvrA2 dissociation has still yet to be demonstrated and that other major elements of the model have been updated. As we see in step 5 of the Sancar model for example

– they didn't know if RNAP left the system at this point, but it was the best guess at the time with the available data; we now know RNAP stays attached to Mfd as part of a DNA-Mfd-RNAP complex which translocates along DNA. This is for instance lost on readers if the authors now decide to claim the "classic model" is correct as they draw in their model Figure 1. Furthermore there is every indication from not just one but a multitude of single-molecule assays (Howan et al. Nature 2012 and Graves et al., NSMB 2016, in addition to Fan et al.) that removal of Mfd-RNAP from DNA is accomplished by UvrA/UvrB. So the authors must negate extensive work to come down so strongly for the "classic" model as their Figure 1 and

overall message convey. This comparison of models is an ineffective narrative through which to explain these findings.

We thank the referee for pointing out this unintentional error. We failed to appreciate that the important insight from the Strick work is not adequately conveyed in the text and that this may be lost on other readers that are less intimately familiar with the work (even though, as we have repeatedly pointed out in this review process, we have found the Strick lab work insightful and instrumental for our own work).

This reviewer is also right in pointing out that our data provide clear evidence for what was merely a guess in 1993. We have therefore chosen to cast our findings in the context of cells as a ‘facilitated dissociation model’.

In light of these comments we have made the following changes:

1. Line 100 – text has now been rephrased to remove reference to the minimal Selby and Sancar model as:
“... these findings lead us to propose a ‘facilitated dissociation’ model for interactions between the transcription repair coupling factor and the repair machineries in cells”
2. We have now removed the reference to the Selby and Sancar model originally in lines 301-302 and replaced it with the ‘facilitated dissociation model’ in lines 285-287.
3. We have now included a panel in Figure 5b that shows this facilitated dissociation model.
4. We have replaced Figure 1b with a simpler model that broadly summarizes the existing models in the field. Corresponding changes in Figure 1 legend.

2. Regarding the nearly full page of new text aimed at the findings of Fan et al., this reviewer absolutely disagrees with the misleadingly formulated statement that “Strick and colleagues failed to detect coordinated handoff” (line 322). The authors cannot ignore that no experiments in Fan et al. were designed to directly test coordinated handoff as no fluorescent UvrA or UvrB were used, so how could they have succeeded or failed at detecting coordinated handoff? If the authors take for proof of this failure the fact that UvrA alone is able to remove Mfd-RNAP, this reviewer notes that the authors are themselves finally acknowledging that UvrA on its own may be able to remove Mfd-RNAP, albeit slowly (see below). If the authors take for proof of this failure that no proteins were observed on the DNA after UvrA/UvrB removal of Mfd-RNAP, that is only because there were no fluorescent UvrA or fluorescent UvrB proteins were used in Fan et al. This straw man representation of other people’s work is unacceptable, and it must be pointed out that this is the umpteenth time in this review process that major errors in the authors’ presentation of Fan et al. have had to be corrected. Although reexamination of prior work is key to the scientific process, such reexamination should not be done in the partial way that the authors have allowed themselves in their reading of Fan et al.

In the interest of clarity for other readers, we have now removed the statements that appear to have confused this referee.

3. In general it is surprising to see the treatment of Fan et al. in this work. Fan et al only showed that UvrAB could provoke release of Mfd-RNAP, and that UvrA could also, albeit more slowly and with reduced efficiency. We note that, as is now acknowledged in this round of review, the authors are also observing this activity of UvrA alone (K37A vs WT). The fact

that Fan et al. observed UvrA releasing Mfd-RNAP without UvrB does not mean it is frequent in vivo where UvrB is present. Again, for Fan et al.'s experiment carried out in the presence of UvrB, the experiment does not directly test coordinated handoff, and the inference in Fan's model that UvrA2B1 was retained on DNA was again a speculation made by the authors (as was done in the 1993 model by Selby and Sancar). Again model comparison is an unattractive means through which this paper could be presented. Is it not preferable to focus on your interesting findings and their biological significance, rather than focus on which model it backs up the most?

We have taken this suggestion on board (see changes in lines 48-51) and now referred to our model for Mfd-UvrAB interactions inside cells as the 'facilitated dissociation model'. See also response to concern#1.

4. The new argument that Fan et al.'s use of 50-100 pM concentration of UvrA leads the authors to "expect that disassembly of the handoff complex via UvrB-facilitated dissociation of Mfd and UvrA would be exceedingly rare" is flawed and certain to mislead readers. The authors state correctly in their point-by-point reply that when present in Fan et al.'s assays UvrB concentration was 250 nM, but this is missing in the manuscript's discussion (line 320). The authors also state correctly that it is widely accepted that UvrA acts as a dimer. Given UvrA's dissociation constant this is practically a necessary assumption for all assays given that there is always a mix of monomer and dimer in everybody's experiments. Low UvrA2 dimer concentration will only affect the time taken for UvrA2/UvrB to bind Mfd-RNAP. This can even be a very slow process given the stability of Mfd-RNAP on DNA. If UvrA does not already come to Mfd-RNAP with UvrB, the time it takes UvrB to join the complex and carry out handoff is only limited by the concentration of Mfd-RNAP, and is independent of the concentration of UvrA. It is therefore unclear why low dimeric UvrA would lead to exceedingly rare handoff via UvrB, apart from the diffusion-based rate-limiting effect of low dimer concentration.

As indicated above, we have removed any comparisons of our results and the lifetime measurements made in Fan et al. Therefore, in the revised text, we no longer include the concentration argument to explain potential differences between ours + Sancar work and the Strick work. In doing so, we disregard R#1 and R#2 comments indicating that this argument should be included in the text.

5. In their rebuttal the authors acknowledge now regarding UvrA(K37A) that "The faster dissociation of Mfd-YPet in the Δ UvrB expressing wild-type UvrA is consistent with a scenario where ATP hydrolysis by UvrA at the proximal site (K37) is able to dissociate the (RNAP)-MfdUvrA2 complex albeit with a rate constant that is approximately ten-fold slower than that observed in the presence of UvrB." This statement, which offers a path for agreement between these data and those of Fan et al., needs to be incorporated into the manuscript's discussion at an appropriate place. This could be accomplished for instance by correcting the text around lines 288-290 to read "...suggests that this ATPase activity alone is sufficient for dissociation of Mfd, albeit at an approximately tenfold slower rate than that observed in the presence of UvrB." It would even be perhaps useful point out that this observation helps reconcile the authors' results with those of Fan et al.

We agree with this suggestion and have now rephrased the text to include this referee's suggestion to enhance the clarity of the text (see highlighted lines 271-273).

6. The correction made line 275 is too vague to be of use to readers and needs to be improved. The authors now write “Whereas the lifetime of the arrested Mfd-UvrA complex has been previously measured to be 15s in vitro, a short-lived complex of Mfd-YPet with a lifetime of 15s was not detected in UvrB- cells. Potentially the two experiments measure different activities.” This reviewer realizes the first sentence introduces confusion between Fan et al.’s use of the term arrested (in which it means that the translocating Mfd-RNAP complex has ceased translocating) and the use the authors make of it for their own experiments (protein localized to DNA rather than diffusing). Of course the assays measure entirely different activities, the authors acknowledged them in their rebuttal but do not clarify here. If the authors insist on making such comparisons, they should correct it here. The bottom line is that the assay of Fan et al. being discussed here does not make any claim as to the amount of time the UvrA-MfdRNAP complex exists on DNA before Mfd-RNAP is arrested in its translocation (present or absent UvrB), and so the time they do measure cannot be held up for comparison to what is observed in this manuscript.

We have taken this referee’s advice to no longer compare or contextualize our measurements with those in Fan et al as these two measurements appear to report on different activities. Therefore, in the interest of clarity this discussion has been entirely revised as below line 261 (highlighted):

“Consistent with previous observations(Fan et al., 2016), our live cell study confirms that the Mfd-UvrA complex is formed at the intersection of the transcription and repair pathways.”

In so doing, we hope to avoid confusing readers regarding the lifetimes measured here and in Fan 2016.

7. Line 366, this reviewer does not understand the authors’ assertion that symmetry-breaking is a problem in the model of Fan et al. On one hand if a UvrA2-UvrB1 complex is deposited on DNA after interaction with Mfd-RNAP then this is precisely a symmetry-breaking complex. In terms of composition it will be stable over longer periods of time the harder it is for a DNABound complex to acquire new subunits (currently not known). On the other hand If the authors’ point is that the Fan model claims UvrA alone to be the TCR taskmaster then there is a misrepresentation of Fan et al. Although Fan et al.’s experiments showed UvrA alone is capable of removing Mfd-RNAP, there are no claims made about how frequent this is in vivo, in particular given the presence of UvrB, and the bottom line of Fan et al has always been that UvrAB is the key to rapidly moving the TCR process towards success.

We have now entirely removed this discussion of symmetry breaking to avoid confusion in the minds of readers.

REVIEWERS' COMMENTS:

Reviewer #3 (Remarks to the Author):

This revised manuscript is compelling and merits publication in Nature Communications. The following are minor points that truly need not be checked in another round of review:

Line 29: "a few exceptions in archaea" but the two references relate to the same organism, so I recommend "a known exception in the archaeon *S. solfataricus*."

Line 69: the comma between *as* and *Mfd* is not needed

Line 83: This is incorrect, it has been argued by these authors that TCR represents a house-keeping DNA repair pathway responsible for maintaining genomic integrity in the absence of SOS induction. Very low levels of exogenous DNA damage need not trigger an SOS-response, which is in fact toxic to the cell if deployed unnecessarily.

Line 84: low levels of exogenous DNA damage need not trigger an SOS-response.

Line 86: The word "TCR" is likely missing between "from" and "becomes".

Line 95: recommend "here we measured" to clarify that this paragraph sets up the story

Line 98: "mutant UvrAB" is unclear. Is it one or the other or both. Recommend "mutant UvrA or UvrB"

Line 99: "are" is not needed here.

Line 263: "of a stable DNA-bound" of is not needed here

REVIEWERS' COMMENTS:

Reviewer #3 (Remarks to the Author):

This revised manuscript is compelling and merits publication in Nature Communications. The following are minor points that truly need not be checked in another round of review:

Line 29: "a few exceptions in archaea" but the two references relate to the same organism, so I recommend "a known exception in the archaeon *S. solfataricus*."

Done.

Line 69: the comma between as and Mfd is not needed

Done.

Line 83: This is incorrect, it has been argued by these authors that TCR represents a house-keeping DNA repair pathway responsible for maintaining genomic integrity in the absence of SOS induction. Very low levels of exogenous DNA damage need not trigger an SOS-response, which is in fact toxic to the cell if deployed unnecessarily.

This idea originates in Fan et al, 2016 and re-iterated in Portman 2018. Our data presented in Ho et al, 2018 are consistent with this idea. We have rephrased this section as:

“The abundant observations of RNAP- and UvrA- dependent interactions of Mfd observed in normal growing cells in our experiments agree with arguments that TCR represents a house-keeping DNA repair pathway.”

Line 84: low levels of exogenous DNA damage need not trigger an SOS-response.

We agree with this statement by the referee.

Line 86: The word "TCR" is likely missing between "from" and "becomes".

Fixed.

Line 95: recommend "here we measured" to clarify that this paragraph sets up the story

Fixed.

Line 98: "mutant UvrAB" is unclear. Is it one or the other or both. Recommend "mutant UvrA or UvrB"

Fixed.

Line 99: "are" is not needed here.

Fixed.

Line 263: "of a stable DNA-bound" of is not needed here

Fixed.